# MODERATE CORESET: A UNIVERSAL METHOD OF DATA SELECTION FOR REAL-WORLD DATA-EFFICIENT DEEP LEARNING

**Xiaobo Xia**[1]    **Jiale Liu**[2]    **Jun Yu**[3]    **Xu Shen**[4]    **Bo Han**[5]    **Tongliang Liu**[1†]
[1]Sydney AI Centre, School of Computer Science, The University of Sydney
[2]School of Computer Science and Technology, Xidian University
[3]Department of Automation, University of Science and Technology of China
[4]Alibaba Group    [5]Department of Computer Science, Hong Kong Baptist University

## ABSTRACT

Deep learning methods nowadays rely on massive data, resulting in substantial costs of data storage and model training. Data selection is a useful tool to alleviate such costs, where a coreset of massive data is extracted to practically perform on par with full data. Based on carefully-designed score criteria, existing methods first count the score of each data point and then select the data points whose scores lie in a certain range to construct a coreset. These methods work well in their respective preconceived scenarios but are not robust to the change of scenarios, since the optimal range of scores varies as the scenario changes. The issue limits the application of these methods, because realistic scenarios often mismatch preconceived ones, and it is inconvenient or unfeasible to tune the criteria and methods accordingly. In this paper, to address the issue, a concept of the moderate coreset is discussed. Specifically, given any score criterion of data selection, different scenarios prefer data points with scores in different intervals. As the score median is a proxy of the score distribution in statistics, the data points with scores close to the score median can be seen as a proxy of full data and generalize different scenarios, which are used to construct the moderate coreset. As a proof-of-concept, a universal method that inherits the moderate coreset and uses the distance of a data point to its class center as the score criterion, is proposed to meet complex realistic scenarios. Extensive experiments confirm the advance of our method over prior state-of-the-art methods, leading to a strong baseline for future research. The implementation is available at https://github.com/tmllab/Moderate-DS.

## 1 INTRODUCTION

Large-scale datasets, comprising millions of examples, are becoming the de-facto standard to achieve state-of-the-art deep learning models (Zhao et al., 2021; Zhao & Bilen, 2021; Yang et al., 2022). Unfortunately, at such scales, both storage costs of the data and computation costs of deep learning model training are tremendous and usually unaffordable by startups or non-profit organizations (Wang et al., 2018a; Coleman et al., 2020; Sorscher et al., 2022; Pooladzandi et al., 2022), which limits the success of deep learning models to specialized equipment and infrastructure (Yang et al., 2023). For instance, the storage needs of ImageNet-22k (Deng et al., 2009) and BDD100K datasets (Yu et al., 2020a) are 1TB and 1.8TB respectively. Training PaLM (Chowdhery et al., 2022) once requires a training dataset containing 780 billion high-quality tokens and then takes almost 1,920 TPU years. Additionally, hyper-parameter tuning or network architecture search could further increase the computation costs, which is pessimistic (Strubell et al., 2019; Schwartz et al., 2020).

Data selection came into being to deal with large data and mitigate the above issues for data-efficient deep learning. More specifically, data selection aims to find the most essential data points and build a coreset of large data. Training on the coreset is expected to continue the model performance achieved by training on the large data (Huang et al., 2021b; Chen et al., 2021). Based on carefully-designed

---

[†]Corresponding author (tongliang.liu@sydney.edu.au).

score criteria, recent works have presented various algorithms of data selection, e.g., in terms of loss values (Han et al., 2018; Jiang et al., 2018), forgetfulness (Toneva et al., 2019; Sorscher et al., 2022), and gradient matching (Paul et al., 2021; Pooladzandi et al., 2022). In respect of procedures, these works first sort the scores achieved by all data points and then simply select the data points with either smaller scores or larger scores, according to different scenarios. For instance, for a loss-based score criterion, if data is presumed a priori to be perfectly labeled, larger-loss data points are more important and selected (Lei et al., 2022). Conversely, if data is corrupted by outliers, smaller-loss data points are more critical because of the concern on model robustness (Lyu & Tsang, 2019).

State-of-the-art methods on data selection can achieve promising performance as they show. However, they are specially designed for preconceived scenarios. The deliberate characteristic makes them work well under certain situations and demands, but not stable or even extremely sensitive to the change of situations or demands, even though the change is slight (the concerns on complex realistic scenarios are detailed in Section 2.2). The issue severely restrict the practical applications of the methods, since realistic scenarios cannot always match preconceived ones well, and realistic demands are frequently changed over time (Hendrycks & Gimpel, 2017; Wu et al., 2021; Arjovsky et al., 2019; Piratla et al., 2020; Creager et al., 2021; Shen et al., 2021; Li et al., 2022b; Wei et al., 2023; Huang et al., 2023). It is inconvenient, troubled, and often unachievable to tweak our method accordingly (Lu et al., 2018).

In this paper, to address the issue, we discuss a new concept about data selection, i.e., the moderate coreset, which is generic in multiple realistic tasks without any task-specific prior knowledge and adjustments. For the construction of the moderate coreset, given any score criterion of data selection, we characterize the score statistics as a distribution with respect to different scenarios. Namely, different scenarios correspond to and require data points with scores in different ranges. The distribution can be generally depicted by the median of scores (James et al., 2013). Accordingly, data points with scores that are close to the score median can be seen as a proxy of all data points, which is used to build a moderate coreset and generalize different scenarios.

As a proof-of-concept, we present a universal method of data selection in complex realistic scenarios. Specifically, working with extracted representations extracted by deep models, we implement the distance of a data point to its class center as a score criterion. Data points with scores close to the score median are selected as a coreset for following tasks. Comparing the complicated and time-consuming data selection procedure of many works, e.g., using Hessian calculation (Yang et al., 2023), our method is simple and does not need to access model architectures and retrain models. We show that existing state-of-the-art methods are not robust to the slight change of presumed scenarios. The proposed method is more superior to them in diverse data selection scenarios.

**Contributions.** Before delving into details, we clearly emphasize our contribution as follows:

- Different from prior works targeting preconceived scenarios, we focus on data selection in the real world, where encountered scenarios always mismatch preconceived ones. The concept of the moderate coreset is proposed to generalize different tasks without any task-specific prior knowledge and fine-tuning.

- As a proof-of-concept, we propose a universal method operating on deep representations of data points for data selection in various realistic scenarios. The method successfully combines the advantages of simplicity and effectiveness.

- Comprehensive experiments for the comparison of our method with state-of-the-arts are provided. Results demonstrate that our method is leading in multiple realistic cases, achieving lower time cost in data selection and better performance on follow-up tasks. This creates a strong baseline of data selection for future research.

## 2 BACKGROUND

### 2.1 DATA SELECTION RECAP

**Data selection vs. data distillation/condensation.** Data selection is a powerful tool as discussed. In data-efficient deep learning, there are also other approaches that are widely studied nowadays, such as data distillation (Cazenavette et al., 2022; Bohdal et al., 2020; Wang et al., 2018a; Such et al., 2020; Nguyen et al., 2021; Sucholutsky & Schonlau, 2021) and data condensation (Wang et al.,

2022; Zhao et al., 2021; Zhao & Bilen, 2021). This series of works focus on *synthesizing* a small but informative dataset as an alternative of the original large-scale dataset. However, the works on data distillation and condensation are criticized for only synthesizing a small number of examples (e.g., 50 images per class) due to computational power limitations (Yang et al., 2023). In addition, their performance is far from satisfactory. Therefore, the performances of data distillation/condensation and data selection are not directly comparable.

**Score criteria in data selection.** Data selection has recently gained lots of interest. As for the procedure of data selection, generally speaking, existing methods first design a score criterion based on a preconceived scenario (e.g., using prior knowledge or other scenario judgment techniques). Afterwards, all data points of a dataset are sorted with the score criterion. The data points with scores in a certain range are selected for follow-up tasks. For instance, regarding the scenario where data is corrupted with outliers, the data points with smaller losses are favored. Popular score criteria include but do not limit to prediction confidence (Pleiss et al., 2020), loss values (Jiang et al., 2018; Wei et al., 2020), margin separation (Har-Peled et al., 2007), gradient norms (Paul et al., 2021), and influence function scores (Koh & Liang, 2017; Yang et al., 2023; Pooladzandi et al., 2022).

## 2.2 Instability of Prior Works to Changed Scenarios

With designed score criteria, prior state-of-the-art methods have achieved promising performance in their respective scenarios. Unfortunately, they are not robust or rather unstable to the change of scenarios, which limits their realistic applications. We detail the issue in three practical respects.

**Distribution shift scenarios.** For methods intended for ideal data, they cannot be applied directly when data contains outliers, because outliers will be identified to contribute more to model learning (Zhang & Sabuncu, 2018) and selected. Conversely, for the methods for the data involving outliers, they will select data points far from the decision boundary for model robustness. However, the uncontaminated data points close to the decision boundary are more important for model performance (Huang et al., 2010), which are entangled with contaminated ones and cannot be identified exactly. A covariate-shift problem (Sugiyama & Kawanabe, 2012) arises to degenerate model performance, if we only use the data points far from the decision boundary for model training.

**Changing demands of coreset sizes.** Different scenarios have different requirements of coreset sizes. Specifically, if a coreset is allowed to have a relatively large size, the coreset built by the data points that contribute more to model generalization (e.g., those close to the decision boundary), can work well for next tasks. However, if a small size is required, the selection with only these data points will make the convergence of deep models difficult to hurt model performance (Bengio et al., 2009; Zhou & Wu, 2021; Sorscher et al., 2022). Namely, the data selection way is unstable to the change in coreset sizes.

**Varied competence on deep models.** Lots of state-of-the-art methods rely on accessing the architectures of deep models and the permission of model retraining for data selection (Toneva et al., 2019; Yang et al., 2023; Sorscher et al., 2022; Paul et al., 2021; Feldman & Zhang, 2020). In some scenarios, requirements for these methods can be met to make them work well. However, if the requirement is not satisfied due to secrecy concerns, these methods will become invalid.

Obviously, the issue of instability to changed scenarios restricts the applications of existing methods, since realistic scenarios cannot always match preconceived ones. Moreover, in many cases, it is expensive or not possible to tune methods frequently according to the changes, which demonstrates the urgency of developing new technologies.

## 3 Methodology

### 3.1 Preliminaries

In the sequel, vectors, matrices, and tuples are denoted by bold-faced letters. We use $\|\cdot\|_p$ as the $\ell_p$ norm of vectors or matrices. Let $[n] = \{1, \ldots, n\}$. Let $\mathbb{I}[\mathcal{B}]$ be the indicator of the event $\mathcal{B}$.

We define the problem of data selection in data-efficient deep learning. Formally, we are given a large-scale dataset $\mathcal{S} = \{\mathbf{s}_1, \ldots, \mathbf{s}_n\}$ with a sample size $n$, where $\mathbf{s}_i = (\mathbf{x}_i, y_i)$, $\mathbf{x}_i \in \mathbb{R}^d$, and $y_i \in [k]$. The aim of data selection here is to find a subset of $\mathcal{S}$ for follow-up tasks, which reduces both storage and training consumption. The subset is called the *coreset* that is expected to practically perform on

par with full data $\mathcal{S}$. We denote the coreset as $\mathcal{S}^* = \{\hat{\mathbf{s}}_1, \ldots, \hat{\mathbf{s}}_m\}$ with a sample size $m$ and $\mathcal{S}^* \subset \mathcal{S}$. The data selection ratio in building the coreset is then $m/n$.

## 3.2 PROCEDURE DESCRIPTION

**Representation extraction.** Given a well-trained deep model denoted by $f(\cdot) = g(h(\cdot))$, where $h(\cdot)$ denotes the part of the model mapping input data to hidden representations at the penultimate layer, and $g(\cdot)$ is the part mapping such hidden representations to the output $f(\cdot)$ for classification. Namely, for a data point $\mathbf{s} = (\mathbf{x}, y)$, its hidden representation is $h(\mathbf{x})$. Therefore, with the trained deep model $f(\cdot)$ and full training data $\mathcal{S} = \{\mathbf{s}_1, \ldots, \mathbf{s}_n\}$, the hidden representations of all data points are acquired as $\{\mathbf{z}_1 = h(\mathbf{x}_1), \ldots, \mathbf{z}_n = h(\mathbf{x}_n)\}$. At the representational level, the class center of each class is

$$\left\{ \mathbf{z}^j = \frac{\sum_{i=1}^n \mathbb{I}[y_i = j] \mathbf{z}_i}{\sum_{i=1}^n \mathbb{I}[y_i = j]} \right\}_{j=1}^k, \tag{1}$$

where the mean of the representations from one class is calculated as the mean of every single dimension in the representations.

**Distance-based score for data selection.** With hidden representations $\{\mathbf{z}_1, \ldots, \mathbf{z}_n\}$ and representational class centers $\{\mathbf{z}^1, \ldots, \mathbf{z}^k\}$, the Euclidean distance from each representation to the corresponding class center can be simply calculated with $d(\mathbf{s}) = \|\mathbf{z} - \mathbf{z}^j\|_2$. The set consisting of the distances is $\{d(\mathbf{s}_1), \ldots, d(\mathbf{s}_n)\}$ with the median as $\mathcal{M}(d(\mathbf{s}))$. All data points are then sorted with an ascending order based on the distance set, which are denoted by $\{d(\tilde{\mathbf{s}}_1), \ldots, d(\tilde{\mathbf{s}}_n)\}$. Afterwards, the data points with distances close to the distance median $\mathcal{M}(d(\mathbf{s}))$ are selected as the coreset $\mathcal{S}^*$, i.e.,

$$\left\{ d(\tilde{\mathbf{s}}_1), \quad \ldots, \quad d(\tilde{\mathbf{s}}_a), \quad d(\tilde{\mathbf{s}}_{a+1}), \quad \ldots, \quad \mathcal{M}(d(\mathbf{s})), \quad \ldots, \quad d(\tilde{\mathbf{s}}_{n-a}), \quad d(\tilde{\mathbf{s}}_{n-a+1}), \quad \ldots, \quad d(\tilde{\mathbf{s}}_n) \right\}$$
$$\downarrow \qquad\qquad \downarrow \qquad\quad \downarrow \qquad\qquad\qquad \downarrow \qquad\qquad\qquad \downarrow \qquad\qquad \downarrow \qquad\qquad\quad \downarrow$$
$$\left\{ \tilde{\mathbf{s}}_1, \quad \ldots, \quad \tilde{\mathbf{s}}_a \quad \tilde{\mathbf{s}}_{a+1}, \quad \ldots, \quad \tilde{\mathbf{s}} \quad \ldots, \quad \tilde{\mathbf{s}}_{n-a}, \quad \tilde{\mathbf{s}}_{n-a+1} \quad \ldots, \quad \tilde{\mathbf{s}}_n \right\}, \tag{2}$$

where $a = (n - m)/2$ relates to the data selection ratio. The set $\{\tilde{\mathbf{s}}_{a+1}, \ldots, \tilde{\mathbf{s}}_{n-a}\}$ is regarded as the coreset $\mathcal{S}^* \subset \mathcal{S}$ for follow-up tasks.

It should be noted that the distance-based score is similar to the confidence-based score. That is to say, for a data point, a closer distance to its class center means that the deep model often gives a higher class posterior probability for this data point. However, the proposed method is easier to apply in the real world. The reason is that representations are more accessible than class posterior probabilities, e.g., directly using the public pretrained models of multiple tasks as representation extractors. Clearly, it is convenient to implement our method without accessing the internal structures of deep models and retraining them. The superiority leads to low time costs of data selection. The lower time costs are consistent with the aim of data-efficient deep learning, which allow our method to be applied to large-scale datasets, e.g., ImageNet (Deng et al., 2009).

## 3.3 MORE JUSTIFICATIONS FOR THE PROPOSED METHOD

We provide more justifications to discuss why our method can work well in realistic scenarios. The justifications come from two perspectives that are popularly used to analyze robustness and generalization of deep models.

**Representation structures.** Prior works have claimed that model performance (e.g., accuracy and robustness) is highly related to representation structures (Yu et al., 2020b; Chan et al., 2022). Good representations should satisfy the following three properties and obtain a trade-off among them. (1) Between-class discriminative: representations of data from different classes are highly uncorrelated. (2) Within-class compressible: representations of data from the same class should be relatively correlated. (3) Maximally diverse representations: variance of representations for each class should be as large as possible as long as they stay uncorrelated from the other classes. Referring to Figure 1, we can see that the proposed method meets the three properties, where the trade-off is achieved. However, the data selection with the other ways cannot balance these three properties simultaneously. This perspective can provide support for the effectiveness of our method in realistic scenarios.

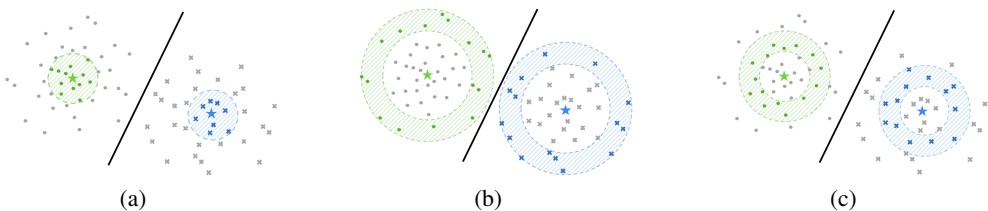

(a)                                     (b)                                     (c)

Figure 1: Illustrations of representation structures achieved by different data selection ways. Different shapes, i.e., the circle and cross, correspond different classes. Pentagrams correspond class centers. The shaded data points are selected for coresets. (a) Selecting data points closer to class centers. (b) Selecting data points far from class centers. (c) Selecting data points with distances close to the distance median.

**Representation information measurement.** Information bottleneck theory (Tishby et al., 2000; Achille & Soatto, 2018) has claimed that optimal representations should be both *sufficient* and *minimal* for various practical tasks. As did in (Achille & Soatto, 2018), we give the following definition.

**Definition 1 (Sufficient and minimal representations)** *We say that a representation $\mathbf{z}$ of $\mathbf{x}$ is sufficient for $y$ if $y \perp\!\!\!\perp \mathbf{x}|\mathbf{z}$ or equivalently if $I(\mathbf{z}; y) = I(\mathbf{x}; y)$; it is minimal when $I(\mathbf{x}; \mathbf{z})$ is smallest among sufficient representations.*

We use a mutual information estimator (Belghazi et al., 2018) to estimate $I(\mathbf{z}; y)$, $I(\mathbf{x}; y)$, and $I(\mathbf{x}; \mathbf{z})$. The technical details of the estimator are provided in Appendix E. Experiments are conducted on CIFAR-100 (Krizhevsky, 2009). Each estimation is repeated 20 times for reported mean.

| Methods / Ratio | 20% | | 30% | |
|---|---|---|---|---|
| Metrics | $\|I(\mathbf{z}; y) - I(\mathbf{x}; y)\|$ | $I(\mathbf{x}; \mathbf{z})$ | $\|I(\mathbf{z}; y) - I(\mathbf{x}; y)\|$ | $I(\mathbf{x}; \mathbf{z})$ |
| Method (a) | 1.901 bits | 2.087 bits | 2.022 bits | 2.384 bits |
| Method (b) | 0.611 bits | 6.512 bits | 1.235 bits | 3.702 bits |
| Method (c) (ours) | 1.652 bits | 4.010 bits | 1.956 bits | 2.737 bits |

Table 1: Comparison of different data selection ways based on representation information measurement. Method (a) selects data points closer to class centers. Method (b) selects data points far from class centers. Method (c) selects data points with the distances close to the distance median.

As shown in Table 1, compared with the other two data selection ways, our method achieves a trade-off between sufficiency and minimality, which justifies the effectiveness of our method.

## 4 EVALUATIONS IN IDEAL SCENARIOS

### 4.1 EXPERIMENTAL SETUP

**Datasets and network structures.** We evaluate the effectiveness of our method on three popularly used datasets, i.e., CIFAR-100 (Krizhevsky, 2009), Tiny-ImageNet (Le & Yang, 2015), and ImageNet-1k (Deng et al., 2009). We first study the effectiveness of data selection methods with a preconfigured network structure. That is to say, the coreset and full data are utilized for the same network structure. ResNet-50 (He et al., 2016) is exploited here. In Appendix D, we provide experiments with multiple network architectures.

**Baselines.** Multiple data selection methods act as baselines for comparison. Specifically, we use (1) Random; (2) Herding (Welling, 2009); (3) Forgetting (Toneva et al., 2019); (4) GraNd-score (Paul et al., 2021); (5) EL2N-score (Paul et al., 2021); (6) Optimization-based (Yang et al., 2023); (7) Self-sup.-selection (Sorscher et al., 2022). Due to the limited page, we provide the technical details of these baselines in Appendix A.

Notice that the methods Forgetting, GraNd-score, and EL2N-score rely on model retraining for data selection. Besides, the method Optimization-based has heavy computational costs, due to the calculation of Hessian in the influence function[1] and its iterative data selection process. The method Self-sup.-selection is troubled by the same issue, where both self-supervised pre-training and cluster-

---

[1] We utilize the PyTorch implementation in https://github.com/nimarb/pytorch_influence_functions.

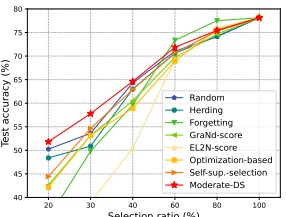 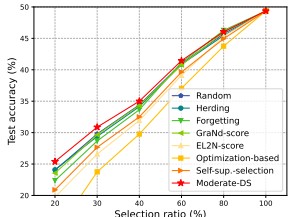 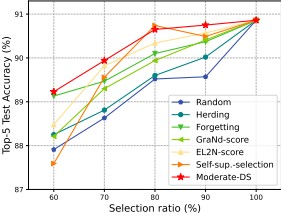

(a) Evaluations on CIFAR-100.    (b) Evaluations on Tiny-ImageNet.    (c) Evaluations on ImageNet-1k.

Figure 2: Illustrations of comparing our proposed method with several data selection baselines on CIFAR-100 (a), Tiny-ImageNet (b), and ImageNet-1k (c). Note that the method Optimization-based is not compared on ImageNet-1k due to its huge time costs of data selection. Exact numerical results can be found in in Appendix B for convenient checks.

ing are time-consuming (Na et al., 2010; Jaiswal et al., 2020). By contrast, Herding and our method only need a distance calculation and a sort operation when there is a representation extractor, leading to low time consumption in data selection. We name the proposed method as **Moderate-DS**.

**Implementation details.** All experiments are conducted on NVIDIA GTX3090 GPUs with Py-Torch (Paszke et al., 2019). For experiments on CIFAR-100, we adopt a batch size of 128, an SGD optimizer with a momentum of 0.9, weight decay of 5e-4, and an initial learning rate of 0.1. The learning rate is divided by 5 after the 60th epoch, the 120th epoch, and the 160th epoch. 200 epochs are set in total. Data augmentation of random crop and random horizontal flip is used. For experiments on Tiny-ImageNet, we adopt a batch size of 256, an SGD optimizer with a momentum of 0.9, weight decay of 1e-4, and an initial learning rate of 0.1. The learning rate is divided by 10 after the 30th epoch and the 60th epoch. 90 epochs are set totally. Random horizontal flips are used to augment training data. In each experiment, we perform five individual runs with different random seeds. For experiments on ImageNet-1k, following (Sorscher et al., 2022), the VISSL library (Goyal et al., 2021) is exploited. We adopt a base learning rate of 0.01, a batch size of 256, an SGD optimizer with a momentum of 0.9, and a weight decay of 1e-3. 100 epochs are set totally. Note that because of huge calculation consumptions, the experiment in each case is performed once. All hyperparameters and experimental settings of training before and after data selection are kept the same.

## 4.2 COMPARISON WITH THE STATE-OF-THE-ARTS

We use test accuracy achieved by training on coresets to verify the effectiveness of the proposed method. For experiments on CIFAR-100 and Tiny-ImageNet, as shown in Figure 2, the proposed method is much competitive with state-of-the-art methods. Particularly, when the data selection ratio is low, e.g., 20%, 30%, and 40%, our method always achieves the best performance. Besides, for experiments on more challenging ImageNet-1k, we can see that our method obtains the best results when data selection ratios are 60%, 70%, and 90%. Also, the achieved result in the 80% case is very close to the best one. Therefore, combining the above discussions on data selection time consumption, we can claim that our method can achieve promising performance on follow-up tasks with lower time costs in data selection.

## 4.3 MORE ANALYSES

**Unseen network structure generalization.** Prior works (Yang et al., 2023) show that, although we perform data selection with a predefined network structure, the selected data, i.e., the coreset, can generalize to those unknown network architectures that are inaccessible during data selection. As did in this line, we train ResNet-50 on CIFAR-100 and Tiny-ImageNet for data selection and further use the selected data to train different network architectures, i.e., SENet (Hu et al., 2018) and EfficientNet-B0 (Tan & Le, 2019). The experimental results in Table 2 show that the selected data by our method has a good generalization on unseen network architectures. The superiority indicates that the proposed method can be employed in a wide range of applications regardless of specific network architectures.

**Ablation study.** We compare the proposed method with the other three data selection ways that share the same distance-based score criterion. Specifically, we perform data selection with (1) data

| CIFAR-100 | | | | | |
|---|---|---|---|---|---|
| Model transfer | ResNet-50→SENet | | ResNet-50→EfficientNet-B0 | | |
| Method / Ratio | 20% | 30% | 20% | 30% | Ave. rank ↓ |
| Random | 53.57±1.18 | 61.08±1.20 | 42.42±1.36 | 53.54±1.11 | 3.8 |
| Herding | 53.77±0.66 | 62.52±0.36 | 43.90±1.10 | 52.75±0.77 | 3.5 |
| Forgetting | 46.35±0.96 | 56.21±0.87 | 37.98±0.15 | 43.89±0.43 | 7.0 |
| GraNd-score | 49.98±0.54 | 57.26±0.74 | 39.34±0.88 | 49.36±1.02 | 5.8 |
| EL2N-score | 39.09±1.32 | 47.31±0.64 | 26.50±0.33 | 37.03±0.52 | 8.0 |
| Optimization-based | 54.02±0.77 | 60.84±0.95 | 43.92±0.50 | 52.88±0.66 | 3.0 |
| Self-sup.-selection | 54.16±0.42 | 60.22±0.55 | 43.65±0.94 | 48.39±1.50 | 4.0 |
| Moderate-DS | **55.57±0.56** | **63.01±0.31** | **48.58±0.15** | **54.37±0.32** | **1.0** |
| Tiny-ImageNet | | | | | |
| Model transfer | ResNet-50→SENet | | ResNet-50→EfficientNet-B0 | | |
| Method / Ratio | 20% | 30% | 20% | 30% | Ave. rank ↓ |
| Random | 34.13±0.71 | 39.57±0.53 | 32.88±1.52 | 39.11±0.94 | 3.5 |
| Herding | 34.86±0.55 | 38.60±0.68 | 32.21±0.54 | 37.53±0.22 | 5.0 |
| Forgetting | 33.40±0.64 | 39.79±0.78 | 31.12±0.21 | 38.38±0.65 | 5.0 |
| GraNd-score | 35.12±0.54 | 41.14±0.42 | 33.20±0.67 | **40.02±0.35** | 1.8 |
| EL2N-score | 31.08±1.11 | 38.26±0.45 | 31.34±0.49 | 36.88±0.32 | 7.5 |
| Optimization-based | 33.18±0.52 | 39.42±0.77 | 32.16±0.90 | 38.52±0.50 | 5.0 |
| Self-sup.-selection | 31.74±0.71 | 38.45±0.39 | 30.99±1.03 | 37.96±0.77 | 7.0 |
| Moderate-DS | **36.04±0.15** | **41.40±0.20** | **34.26±0.48** | 39.57±0.29 | **1.3** |

Table 2: Mean and standard deviation of results (%) on CIFAR-100 and Tiny-ImageNet with transferred models. The average rank is calculated with the ranks of a method in 20% and 30% cases of "ResNet-50→SENet" and "ResNet-50→EfficientNet-B0". The best result in each case is in bold.

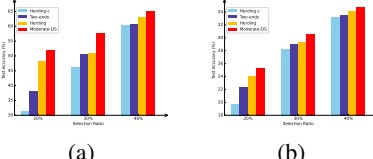

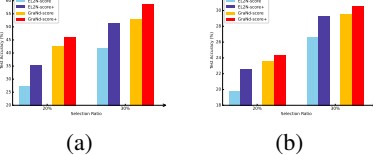

| (a) | (b) | | (a) | (b) |

Figure 3: Illustrations of results of ablation study on different data selection ways. (a) Evaluations on CIFAR-100. (b) Evaluations on Tiny-ImageNet.

Figure 4: Illustrations of boosting advanced methods with moderate coresets. (a) Evaluations on CIFAR-100. (b) Evaluations on Tiny-ImageNet.

points close to class centers (i.e., Herding); (2) data points far from class centers (named as Herding-c); (3) data points at two ends, i.e., both data points close to class centers and far from class centers (named as Two-ends). Experiments are conducted on CIFAR-100 and Tiny-ImageNet. Note that data points far from class centers make model training difficult, hence degenerating model performance, especially when the coreset is limited to a small size. The results are shown in Figure 3, which demonstrate that our method is more promising than other data selection ways.

**Boosting baselines with moderate coresets.** As discussed before, we propose the concept of moderate coresets, which can be applied to other score-based data selection methods. To support the claim, we apply moderate coresets to boost baselines GraNd-score and EL2N-score. In other words, based on the score criteria of GraNd-score and EL2N-score, we select the data points with scores close to the score median. We name the boosted methods as GraNd-score+ and EL2N-score+ respectively. Experiments are conducted on CIFAR-100 and Tiny-ImageNet. We provide the evidence in Figure 4, which demonstrates the use of moderate coresets can bring clear performance improvement.

## 5 EVALUATIONS IN COMPLEX REALISTIC SCENARIOS

### 5.1 ROBUSTNESS TO CORRUPTED IMAGES

In realistic scenes, training data is always polluted by corrupted images (Wang et al., 2018b; Hendrycks & Dietterich, 2019; Li et al., 2021; Xia et al., 2021b). Here, we demonstrate that our method is more robust than several state-of-the-art methods, when corrupted images occur. Specifi-

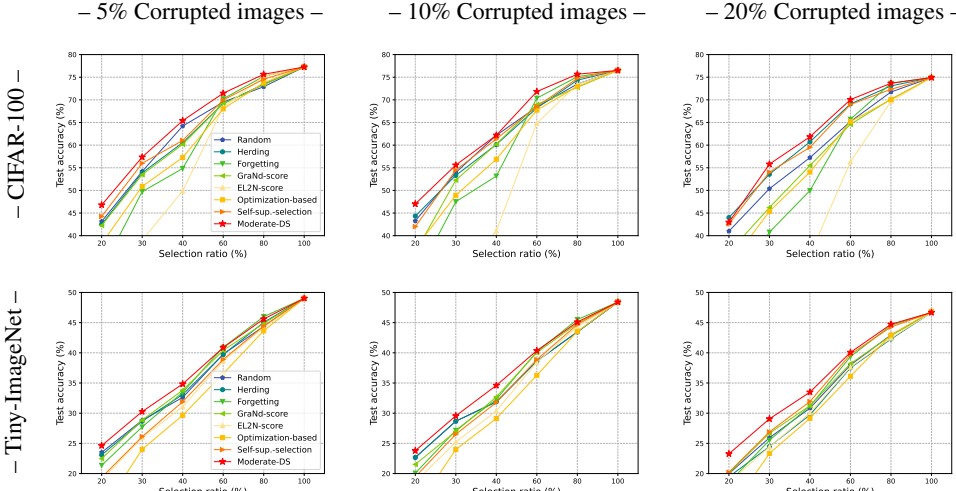

Figure 5: Illustrations of comparing our proposed method with several data selection baselines on synthetic CIFAR-100 and Tiny-ImageNet, where corruption is injected. Exact numerical results (mean±standard deviation) can be found in in Appendix B for convenient checks.

cally, to corrupt images, we employ the following five types of realistic noises, i.e., *Gaussion noise*, *random occlusion*, *resolution*, *fog*, and *motion blur*, as shown in Figure 6 of Appendix C. The noises are then injected into the fractional images of CIFAR-100 and Tiny-ImageNet. The other settings are kept unchanged. The corruption rate is set to 5%, 10%, and 20% respectively. Experimental results are presented in Figure 5.

As can be seen, for CIFAR-100 and Tiny-ImageNet with corrupted images, with the variability of selection ratios, our method outperforms baseline methods in most cases. More specifically, for the method that prefers easier data to reduce the side-effect of corrupted images, e.g., Herding, although corrupted images can be less selected, a covariate-shift problem (Sugiyama & Kawanabe, 2012) arises to degenerate model performance if we only use these data. In addition, for the method that prefers harder data, e.g., Forgetting, the corruptions are incorrectly selected, leading to unsatisfactory performance. Compared with baselines, our method selects moderate data to avoid corruption meanwhile ensure generalization, which hence achieves better performance.

## 5.2 ROBUSTNESS TO LABEL NOISE

Real-world datasets inevitably involve label noise, where partial clean labels are flipped to incorrect ones, resulting in mislabeled data (Northcutt et al., 2021; Xia et al., 2019; 2020). Label-noise robustness is of significance for the method's application (Song et al., 2022; Wei & Liu, 2021; Zhu et al., 2022). To discuss the robustness of different data selection methods against label noise, we conduct synthetic experiments with CIFAR-100 and Tiny-ImageNet. Symmetric label noise (Patrini et al., 2017; Xia et al., 2021a; Li et al., 2022a) is injected to generate mislabeled data. The noise rate is set to 20%. Although mislabeled data makes class centers somewhat biased, Herding and our method can empirically works well. Experimental results are provided in Table 3. The results demonstrate that, compared with prior state-of-the-arts, our method is very effective in the situation of data selection under label noise.

## 5.3 DEFENSE AGAINST ADVERSARIAL ATTACKS

It has been shown that deep networks are vulnerable to adversarial examples that are crafted by adding imperceptible but adversarial noise on natural examples (Szegedy et al., 2014; Ma et al., 2018b; Eykholt et al., 2018; Huang et al., 2021a; Zhou et al., 2021). In the real world, adversarial robustness is greatly important for the application of a method (Tramèr et al., 2018; Dong et al., 2022; Zhou et al., 2022).

We utilize two popularly used adversarial attacks, i.e., PGD attacks (Madry et al., 2017) and GS attacks (Goodfellow et al., 2014). We set default perturbation budget $\epsilon = 8/255$ for both CIFAR-100

| Method / Ratio | CIFAR-100 (Label noise) | | Tiny-ImageNet (Label noise) | | |
|---|---|---|---|---|---|
| | 20% | 30% | 20% | 30% | Ave. rank ↓ |
| Random | 34.47±0.64 | 43.26±1.21 | 17.78±0.44 | 23.88±0.42 | 3.8 |
| Herding | **42.29±1.75** | **50.52±3.38** | 18.98±0.44 | 24.23±0.29 | 1.5 |
| Forgetting | 36.53±1.11 | 45.78±1.04 | 13.20±0.38 | 21.79±0.43 | 5.0 |
| GraNd-score | 31.72±0.67 | 42.80±0.30 | 18.28±0.32 | 23.72±0.18 | 4.5 |
| EL2N-score | 29.82±1.19 | 33.62±2.35 | 13.93±0.69 | 18.57±0.31 | 7.8 |
| Optimization-based | 32.79±0.62 | 41.80±1.69 | 14.77±0.95 | 22.52±0.77 | 5.8 |
| Self-sup.-selection | 31.08±0.78 | 41.87±0.63 | 15.10±0.73 | 21.01±0.36 | 6.3 |
| Moderate-DS | 40.25±0.12 | 48.53±1.60 | **19.64±0.40** | **24.96±0.30** | **1.5** |

Table 3: Mean and standard deviation of experimental results (%) on CIFAR-100 and Tiny-ImageNet with mislabeled data. The average rank is calculated with the ranks of a method in 20% and 30% cases in two synthetic datasets. The best result in each case is in bold.

| Method / Ratio | CIFAR-100 (PGD attacks) | | CIFAR-100 (GS attacks) | | |
|---|---|---|---|---|---|
| | 20% | 30% | 20% | 30% | Ave. rank ↓ |
| Random | 43.23±0.31 | **52.86±0.34** | 44.23±0.41 | 53.44±0.44 | 1.8 |
| Herding | 40.21±0.72 | 49.62±0.65 | 39.92±1.03 | 50.14±0.15 | 5.0 |
| Forgetting | 35.90±1.30 | 47.37±0.99 | 37.55±0.53 | 46.88±1.91 | 6.8 |
| GraNd-score | 40.87±0.84 | 50.13±0.30 | 40.77±1.11 | 49.88±0.83 | 3.8 |
| EL2N-score | 26.61±0.58 | 34.50±1.02 | 26.72±0.66 | 35.55±1.30 | 8.0 |
| Optimization-based | 38.29±1.77 | 46.25±1.82 | 41.36±0.92 | 49.10±0.81 | 5.5 |
| Self-sup.-selection | 40.53±1.15 | 49.95±0.50 | 40.74±1.66 | 51.23±0.25 | 4.0 |
| Moderate-DS | **43.60±0.97** | 51.66±0.39 | **44.69±0.68** | **53.71±0.37** | **1.3** |
| | **Tiny-ImageNet (PGD attacks)** | | **Tiny-ImageNet (GS attacks)** | | |
| Method / Ratio | 20% | 30% | 20% | 30% | Ave. rank ↓ |
| Random | 20.93±0.30 | 26.60±0.98 | 22.43±0.31 | 26.89±0.31 | 3.5 |
| Herding | 21.61±0.36 | 25.95±0.19 | 23.04±0.28 | 27.39±0.14 | 3.0 |
| Forgetting | 20.38±0.47 | 26.12±0.19 | 22.06±0.31 | 27.21±0.21 | 4.5 |
| GraNd-score | 20.76±0.21 | 26.34±0.32 | 22.56±0.30 | 27.52±0.40 | 3.0 |
| EL2N-score | 16.67±0.62 | 22.36±0.42 | 19.93±0.57 | 24.65±0.32 | 7.8 |
| Optimization-based | 19.26±0.77 | 24.55±0.92 | 21.26±0.24 | 25.88±0.37 | 6.0 |
| Self-sup.-selection | 19.23±0.46 | 23.92±0.51 | 19.70±0.20 | 24.73±0.39 | 7.3 |
| Moderate-DS | **21.81±0.37** | **27.11±0.20** | **23.20±0.13** | **28.89±0.27** | **1.0** |

Table 4: Mean and standard deviation of experimental results (%) on CIFAR-100 and Tiny-ImageNet with adversarial examples. The average rank is calculated with the ranks of a method in 20% and 30% cases of two types of attacks. The best result in each case is in bold.

and Tiny-ImageNet. We use the adversarial attacks and the models trained on CIFAR-100 and Tiny-ImageNet to achieve adversarial examples. Afterwards, different methods are applied on adversarial examples and model retraining on selected data. Experimental results are provided in Table 4. As can be seen, the proposed method is more robust than several state-of-the-art methods, leading to the first average rank. The results confirm that our method is more competitive to be applied in practice.

## 6 CONCLUSION

In this paper, we focus on data selection to boost data-efficient deep learning. Different from existing works that are usually limited to preconceived scenarios, we propose a concept of the moderate coreset to generalize different scenarios. As a proof-of-concept, a universal method operating with data representations is presented for data selection in various circumstances. Extensive experiments confirm the effectiveness of the proposed method. For future work, we are interested in adapting our method to other domains such as natural language processing. Furthermore, we are also interested in applying the concept of the moderate coreset concept to multiple advanced data selection methods theoretically and empirically.

ACKNOWLEDGEMENTS

Xiaobo Xia was supported by Australian Research Council Project DE-190101473 and Google PhD Fellowship. Jun Yu is sponsored by Natural Science Foundation of China (62276242), CAAI-Huawei MindSpore Open Fund (CAAIXSJLJJ-2021-016B, CAAIXSJLJJ-2022-001A), An-hui Province Key Research and Development Program (202104a05020007), USTC-IAT Application Sci. & Tech. Achievement Cultivation Program (JL06521001Y). Xu Shen was (partially) supported by the National Key R&D Program of China under Grant 2020AAA0103902. Bo Han was supported by NSFC Young Scientists Fund No. 62006202, Guangdong Basic and Applied Basic Research Foundation No. 2022A1515011652 and RGC Early Career Scheme No. 22200720. Tongliang Liu was partially supported by Australian Research Council Projects IC-190100031, LP-220100527, DP-220102121, and FT-220100318. The authors would give special thanks to Wenhao Yang from Nanjing University for helpful discussions.

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

## A    TECHNICAL DETAILS OF BASELINE METHODS

Here we introduce the technical details of baselines in the following.

- "Random" means that we randomly select partial data from full data.
- "Herding" (Welling, 2009) selects data points that are closer to class centers.
- "Forgetting" (Toneva et al., 2019) selects data points that are easy to be forgotten during optimization.
- "GraNd-score" (Paul et al., 2021) includes data points with larger loss gradient norms.
- "EL2N-score" (Paul et al., 2021) focuses on data points with larger norms of the error vector that is the predicted class probabilities minus one-hot label encoding.
- "Optimization-based" (Yang et al., 2023) employs the influence function (Koh & Liang, 2017) and picks data points that yields strictly constrained generalization gap.
- "Self-sup.-selection" (Sorscher et al., 2022) selects data points on the difficulty of each data point by the distance to its nearest cluster centroid, after self-supervised pre-training and clustering. To avoid the tuning of the cluster number, we set it to the class number. Data points with larger distances are selected here.

## B    EXACT NUMERICAL EXPERIMENTAL RESULTS

In the main paper, we have provided the illustrations of comparing the proposed method (Moderate-DS) with several state-of-the-arts. Here, exact numerical experimental results are presented in Tables 5, 6, and 7 for checks and references.

## C    SUPPLEMENTARY EXPERIMENTAL SETTINGS

In the main paper, we employ five types of realistic noises, i.e., *Gaussion noise*, *random occlusion*, *resolution*, *fog*, and *motion blur* to corrupt images. The noise is shown in Figure 6.

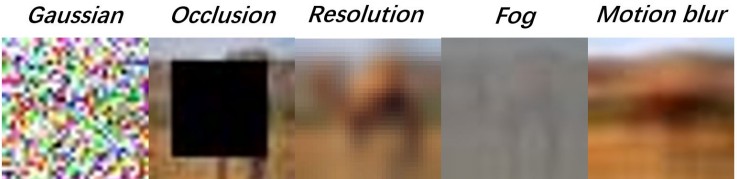

*Gaussian*    *Occlusion*    *Resolution*    *Fog*    *Motion blur*

Figure 6: Examples of noise injected to CIFAR-100 images.

## D    EXPERIMENTS WITH DIFFERENT NETWORK ARCHITECTURES

### D.1    EXPERIMENTS WITH DIFFERENT CONVOLUTIONAL NEURAL NETWORKS

To verify the effectiveness of our method with different convolutional neural networks (CNNs), we employ VGG-16 (Simonyan & Zisserman, 2014) and ShuffleNet V2 (Ma et al., 2018a). Experiments are conducted on Tiny-ImageNet. Results are provided in Table 8. As can be seen, with the changes of different network architectures, our Moderate-DS still works very well.

### D.2    EXPERIMENTS WITH TRANSFORMER

To further demonstrate the superiority of our Moderate-DS, we employ Transformer. The implementation is based on the public Github repository[2], where ViT small is used. We conduct experiments on simulated CIFAR-100. Following the main paper, we consider: (1) experiments with original datasets (C); (2) experiments with 20% corrupted images (C+C); (3) experiments with 20% mislabeled data (C+M); (4) experiments with adversiral examples (C+A). We consider a more challenging

---

[2]https://github.com/kentaroy47/vision-transformers-cifar10

| CIFAR-100 | | | | | | | |
|---|---|---|---|---|---|---|---|
| Method / Selection ratio | 20% | 30% | 40% | 60% | 80% | 100% | Ave. rank ↓ |
| Random | 50.26±3.24 | 53.61±2.73 | 64.32±1.77 | 71.03±0.75 | 74.12±0.56 | 78.14±0.55 | 3.6 |
| Herding | 48.39±1.42 | 50.89±0.97 | 62.99±0.61 | 70.61±0.44 | 74.21±0.49 | 78.14±0.55 | 4.8 |
| Forgetting | 35.57±1.40 | 49.83±0.91 | 59.65±2.50 | **73.34±0.86** | **77.50±0.53** | 78.14±0.55 | 4.4 |
| GraNd-score | 42.65±1.39 | 53.14±1.28 | 60.52±0.79 | 69.70±0.68 | 74.67±0.79 | 78.14±0.55 | 5.4 |
| EL2N-score | 27.32±1.16 | 41.98±0.54 | 50.47±1.20 | 69.23±1.00 | 75.96±0.88 | 78.14±0.55 | 6.6 |
| Optimization-based | 42.16±3.30 | 53.19±2.14 | 58.93±0.98 | 68.93±0.70 | 75.62±0.33 | 78.14±0.55 | 5.6 |
| Self-sup.-selection | 44.45±2.51 | 54.63±2.10 | 62.91±1.20 | 70.70±0.82 | 75.29±0.45 | 78.14±0.55 | 3.8 |
| Moderate-DS | **51.83±0.52** | **57.79±1.61** | **64.92±0.93** | 71.87±0.91 | 75.44±0.40 | 78.14±0.55 | **1.8** |
| CIFAR-100 with 5% corrupted images | | | | | | | |
| Method / Selection ratio | 20% | 30% | 40% | 60% | 80% | 100% | Ave. rank ↓ |
| Random | 43.14±3.04 | 54.19±2.92 | 64.21±2.39 | 69.50±1.06 | 72.90±0.52 | 77.26±0.39 | 4.0 |
| Herding | 42.50±1.27 | 53.88±0.37 | 60.54±0.94 | 69.15±0.55 | 73.47±0.89 | 77.26±0.39 | 4.8 |
| Forgetting | 32.42±0.74 | 49.72±1.64 | 54.84±2.20 | 70.22±2.00 | 75.19±0.40 | 77.26±0.39 | 5.0 |
| GraNd-score | 42.24±0.57 | 53.48±0.76 | 60.17±1.66 | 69.16±0.81 | 73.35±0.81 | 77.26±0.39 | 5.4 |
| EL2N-score | 26.13±1.75 | 39.01±2.14 | 49.89±1.87 | 68.36±1.41 | 75.18±0.36 | 77.26±0.39 | 6.8 |
| Optimization-based | 38.25±3.04 | 50.88±0.62 | 57.26±0.93 | 68.02±0.39 | 73.77±0.56 | 77.26±0.39 | 6.2 |
| Self-sup.-selection | 44.24±0.48 | 55.99±1.21 | 61.03±0.59 | 69.96±1.07 | 74.56±1.17 | 77.26±0.39 | 2.8 |
| Moderate-DS | **46.78±1.90** | **57.36±1.22** | **65.40±0.39** | **71.46±0.19** | **75.64±0.61** | 77.26±0.39 | **1.0** |
| CIFAR-100 with 10% corrupted images | | | | | | | |
| Method / Selection ratio | 20% | 30% | 40% | 60% | 80% | 100% | Ave. rank ↓ |
| Random | 43.27±3.01 | 53.94±2.78 | 62.17±1.29 | 68.41±1.21 | 73.50±0.73 | 76.50±0.63 | 3.6 |
| Herding | 44.34±1.07 | 53.31±1.49 | 60.13±0.38 | 68.20±0.74 | 74.34±0.14 | 76.50±0.63 | 4.2 |
| Forgetting | 30.43±0.70 | 47.50±1.43 | 53.16±0.44 | 70.36±0.82 | 75.11±0.71 | 76.50±0.63 | 5.0 |
| GraNd-score | 36.36±1.06 | 52.26±2.06 | 60.22±1.39 | 68.96±0.68 | 72.78±0.51 | 76.50±0.63 | 5.2 |
| EL2N-score | 21.75±1.56 | 30.80±2.23 | 41.06±1.23 | 64.82±1.48 | 73.47±1.30 | 76.50±0.63 | 7.6 |
| Optimization-based | 37.22±0.39 | 48.92±1.38 | 56.88±1.48 | 67.73±2.15 | 72.94±1.90 | 76.50±0.63 | 6.2 |
| Self-sup.-selection | 42.01±1.31 | 54.47±1.19 | 61.37±0.68 | 68.52±1.24 | 74.73±0.36 | 76.50±0.63 | 3.2 |
| Moderate-DS | **47.02±0.66** | **55.60±1.67** | **62.18±1.86** | **71.83±0.78** | **75.66±0.66** | 76.50±0.63 | **1.0** |
| CIFAR-100 with 20% corrupted images | | | | | | | |
| Method / Selection ratio | 20% | 30% | 40% | 60% | 80% | 100% | Ave. rank ↓ |
| Random | 40.99±1.46 | 50.38±1.39 | 57.24±0.65 | 65.21±1.31 | 71.74±0.28 | 74.92±0.88 | 4.2 |
| Herding | **44.42±0.46** | 53.57±0.31 | 60.72±1.78 | 69.09±1.73 | 73.08±0.98 | 74.92±0.88 | 2.0 |
| Forgetting | 26.39±0.17 | 40.78±2.02 | 49.95±2.31 | 65.71±1.28 | **73.67±1.12** | 74.92±0.88 | 5.2 |
| GraNd-score | 36.33±2.66 | 46.21±1.48 | 55.51±0.76 | 64.59±2.04 | 70.14±1.36 | 74.92±0.88 | 5.4 |
| EL2N-score | 21.64±2.03 | 23.78±1.66 | 35.71±1.17 | 56.32±0.86 | 69.66±0.43 | 74.92±0.88 | 4.6 |
| Optimization-based | 33.42±1.60 | 45.37±2.81 | 54.06±1.74 | 65.19±1.27 | 70.06±0.83 | 74.92±0.88 | 6.0 |
| Self-sup.-selection | 42.61±2.44 | 54.04±1.90 | 59.51±1.22 | 68.97±0.96 | 72.33±0.20 | 74.92±0.88 | 2.8 |
| Moderate-DS | 42.98±0.87 | **55.80±0.95** | **61.84±1.96** | **70.05±1.29** | 73.67±0.30 | 74.92±0.88 | **1.2** |

Table 5: Mean and standard deviation of experimental results (%) on different versions of CIFAR-100. The average rank is calculated with the ranks of a method in 20%, 30%, 40%, 60%, and 80% cases. The best result in each case is in bold.

setting, where the selection ratio is 20%. Empirical results in Table 9 verify the effectiveness of our method with Transformer.

### D.3 Supplementary Experiments with Mislabeled Data

In the main paper, we have demonstrated the effectiveness of our method, when the noise rate of 20%. Here, we improve the noise rate to 35% to demonstrate the superiority of our method Moderate-DS. We provide results in Table 10.

Besides, we before set the selection ratio to 20% and 30% when mislabeled data occur. We furthermore increase the selection ratio to show the advantage of our method. Experiments are conducted on Tiny-ImageNet, where results are provided in Table 11 and Figure 7. As we can see, although the baseline Forgetting achieves competitive performance in some cases, our method works well in most cases. When the selection ratio is small (20%, 30%, and 40%), the data-selection task is more challenging. Our method can achieve the best performance. Besides, the average rank achieved by our method is the best, which confirms the superiority of Moderate-DS.

### D.4 Supplementary Experiments with Corrupted Images

We increase the ratio of corrupted images to 30% to show the effectiveness of the proposed method. Experimental results are provided in Table 12. Under this experimental setting, Herding is strong. Compared to it, Moderate is more powerful, which achieves the best average rank.

| Tiny-ImageNet | | | | | | | |
|---|---|---|---|---|---|---|---|
| Method / Selection ratio | 20% | 30% | 40% | 60% | 80% | 100% | Ave. rank ↓ |
| Random | 24.02±0.41 | 29.79±0.27 | 34.41±0.46 | 40.96±0.47 | 45.74±0.61 | 49.36±0.25 | 2.8 |
| Herding | 24.09±0.45 | 29.39±0.53 | 34.13±0.37 | 40.86±0.61 | 45.45±0.33 | 49.36±0.25 | 3.8 |
| Forgetting | 22.37±0.71 | 28.67±0.54 | 33.64±0.32 | 41.14±0.43 | **46.77±0.31** | 49.36±0.25 | 3.6 |
| GraNd-score | 23.56±0.52 | 29.66±0.37 | 34.33±0.50 | 40.77±0.42 | 45.96±0.56 | 49.36±0.25 | 3.6 |
| EL2N-score | 19.74±0.26 | 26.58±0.40 | 31.93±0.28 | 39.12±0.46 | 45.32±0.27 | 49.36±0.25 | 6.8 |
| Optimization-based | 13.88±2.17 | 23.75±1.62 | 29.77±0.94 | 37.05±2.81 | 43.76±1.50 | 49.36±0.25 | 8.0 |
| Self-sup.-selection | 20.89±0.42 | 27.66±0.50 | 32.50±0.30 | 39.64±0.39 | 44.94±0.34 | 49.36±0.25 | 6.2 |
| Moderate-DS | **25.29±0.38** | **30.57±0.20** | **34.81±0.51** | **41.45±0.44** | 46.06±0.33 | 49.36±0.25 | **1.2** |
| Tiny-ImageNet with 5% corrupted images | | | | | | | |
| Method / Selection ratio | 20% | 30% | 40% | 60% | 80% | 100% | Ave. rank ↓ |
| Random | 23.51±0.22 | 28.82±0.72 | 32.61±0.68 | 39.77±0.35 | 44.37±0.34 | 49.02±0.35 | 4.0 |
| Herding | 23.09±0.53 | 28.67±0.37 | 33.09±0.32 | 39.71±0.31 | 45.04±0.15 | 49.02±0.35 | 3.8 |
| Forgetting | 21.36±0.28 | 27.72±0.43 | 33.45±0.21 | **40.92±0.45** | **45.99±0.51** | 49.02±0.35 | 3.0 |
| GraNd-score | 22.47±0.23 | 28.85±0.83 | 33.81±0.24 | 40.40±0.15 | 44.86±0.49 | 49.02±0.35 | 3.0 |
| EL2N-score | 18.98±0.72 | 25.96±0.28 | 31.07±0.63 | 38.65±0.36 | 44.21±0.68 | 49.02±0.35 | 7.0 |
| Optimization-based | 13.65±1.26 | 24.02±1.35 | 29.65±1.86 | 36.55±1.84 | 43.64±0.71 | 49.02±0.35 | 8.0 |
| Self-sup.-selection | 19.35±0.57 | 26.11±0.31 | 31.90±0.37 | 38.91±0.29 | 44.43±0.42 | 49.02±0.35 | 5.8 |
| Moderate-DS | **24.63±0.78** | **30.27±0.16** | **34.84±0.24** | 40.86±0.42 | 45.60±0.31 | 49.02±0.35 | **1.4** |
| Tiny-ImageNet with 10% corrupted images | | | | | | | |
| Method / Selection ratio | 20% | 30% | 40% | 60% | 80% | 100% | Ave. rank ↓ |
| Random | 22.67±0.27 | 28.67±0.52 | 31.88±0.30 | 38.63±0.36 | 43.46±0.20 | 48.40±0.32 | 4.4 |
| Herding | 22.01±0.18 | 27.82±0.11 | 31.82±0.26 | 39.37±0.18 | 44.18±0.27 | 48.40±0.32 | 4.2 |
| Forgetting | 20.06±0.48 | 27.17±0.36 | 32.31±0.22 | 40.19±0.29 | **45.51±0.48** | 48.40±0.32 | 3.0 |
| GraNd-score | 21.52±0.48 | 26.98±0.43 | 32.70±0.19 | 40.03±0.26 | 44.87±0.35 | 48.40±0.32 | 3.4 |
| EL2N-score | 18.59±0.13 | 25.23±0.18 | 30.37±0.22 | 38.44±0.32 | 44.32±1.07 | 48.40±0.32 | 6.6 |
| Optimization-based | 14.05±1.74 | 23.98±1.77 | 29.12±0.61 | 36.28±1.88 | 43.52±0.31 | 48.40±0.32 | 7.8 |
| Self-sup.-selection | 19.47±0.26 | 26.51±0.55 | 31.78±0.14 | 38.87±0.54 | 44.69±0.29 | 48.40±0.32 | 5.4 |
| Moderate-DS | **23.79±0.16** | **29.56±0.16** | **34.60±0.12** | **40.36±0.27** | 45.10±0.23 | 48.40±0.32 | **1.2** |
| Tiny-ImageNet with 20% corrupted images | | | | | | | |
| Method / Selection ratio | 20% | 30% | 40% | 60% | 80% | 100% | Ave. rank ↓ |
| Random | 19.99±0.42 | 25.93±0.53 | 30.83±0.44 | 37.98±0.31 | 42.96±0.62 | 46.68±0.43 | 4.4 |
| Herding | 19.46±0.14 | 24.47±0.33 | 29.72±0.39 | 37.50±0.59 | 42.28±0.30 | 46.68±0.43 | 6.6 |
| Forgetting | 18.47±0.46 | 25.53±0.23 | 31.17±0.24 | 39.35±0.44 | 44.55±0.67 | 46.68±0.43 | 4.2 |
| GraNd-score | 20.07±0.49 | 26.68±0.44 | 31.25±0.40 | 38.21±0.49 | 42.84±0.72 | 46.68±0.43 | 1.8 |
| EL2N-score | 18.57±0.30 | 24.42±0.44 | 30.04±0.15 | 37.62±0.44 | 42.43±0.61 | 46.68±0.43 | 6.4 |
| Optimization-based | 13.71±0.26 | 23.33±1.84 | 29.15±2.84 | 36.12±1.86 | 42.94±0.52 | 46.88±0.43 | 7.4 |
| Self-sup.-selection | 20.22±0.23 | 26.90±0.50 | 31.93±0.49 | 39.74±0.52 | 44.27±0.10 | 46.68±0.43 | 2.2 |
| Moderate-DS | **23.27±0.33** | **29.06±0.36** | **33.48±0.11** | **40.07±0.36** | **44.73±0.39** | 46.68±0.43 | **1.0** |

Table 6: Mean and standard deviation of experimental results (%) on Tiny-ImageNet. The average rank is calculated with the ranks of a method in 20%, 30%, 40%, 60%, and 80% cases. The best result in each case is in bold.

| ImageNet-1k | | | | | | |
|---|---|---|---|---|---|---|
| Method / Selection ratio | 60% | 70% | 80% | 90% | 100% | Ave. rank ↓ |
| Random | 87.91 | 88.63 | 89.52 | 89.57 | 90.86 | 6.8 |
| Herding | 88.25 | 88.81 | 89.60 | 90.02 | 90.86 | 5.5 |
| Forgetting | 89.13 | 89.47 | 90.10 | 90.37 | 90.86 | 3.8 |
| GraNd-score | 88.21 | 89.30 | 89.94 | 90.41 | 90.86 | 4.8 |
| EL2N-score | 88.48 | 89.82 | 90.34 | 90.57 | 90.86 | 2.5 |
| Optimization-based | - | - | - | - | - | - |
| Self-sup.-selection | 87.59 | 89.56 | **90.74** | 90.49 | 90.86 | 3.5 |
| Moderate-DS | **89.23** | **89.94** | 90.65 | **90.75** | 90.86 | **1.3** |

Table 7: Top-5 test accuracy (%) on ImageNet-1k. The average rank is calculated with the ranks of a method in 60%, 70%, 80%, and 90% cases. The best result in each case is in bold.

## D.5 SUPPLEMENTARY EXPERIMENTS WITH SIMULTANEOUS CORRUPTED, MISLABELED, AND ADVERSARIAL EXAMPLES

In the main paper, we have demonstrated the effectiveness of our method, when training data are polluted by corrupted images, mislabeled data, and adversarial attacks respectively. Here, we further present the experiments with *simultaneous* corrupted, mislabeled, and adversarial examples. The ratio of each is 10%. Experimental results are provided in Table 13, which clearly verify the effectiveness of the proposed method.

| Tiny-ImageNet | | | | | |
|---|---|---|---|---|---|
| Method / Selection ratio | 20% (V) | 30% (V) | 20% (S) | 30% (S) | Ave. rank ↓ |
| Random | 29.63±0.43 | 35.38±0.83 | 32.40±1.06 | 39.13±0.81 | 3.5 |
| Herding | 31.05±0.22 | 36.27±0.57 | 33.10±0.39 | 38.65±0.22 | 2.3 |
| Forgetting | 27.53±0.36 | 35.61±0.39 | 27.82±0.56 | 36.26±0.51 | 5.8 |
| GraNd-score | 29.93±0.95 | 35.61±0.39 | 29.56±0.46 | 37.40±0.38 | 4.3 |
| EL2N-score | 26.47±0.31 | 33.19±0.51 | 28.18±0.27 | 35.81±0.29 | 7.3 |
| Optimization-based | 25.92±0.64 | 34.82±1.29 | 31.37±1.14 | 38.22±0.78 | 5.3 |
| Self-sup.-selection | 25.16±1.10 | 33.30±0.94 | 29.47±0.56 | 36.68±0.36 | 6.8 |
| Moderate-DS | **31.45±0.32** | **37.89±0.36** | **33.32±0.41** | **39.68±0.34** | **1.0** |
| Tiny-ImageNet with 20% corrupted images | | | | | |
| Method / Selection ratio | 20% (V) | 30% (V) | 20% (S) | 30% (S) | Ave. rank ↓ |
| Random | 26.33±0.88 | 31.57±1.31 | 29.15±0.83 | 34.72±1.00 | 2.5 |
| Herding | 18.03±0.33 | 25.77±0.34 | 23.33±0.43 | 31.73±0.38 | 7.0 |
| Forgetting | 19.41±0.57 | 28.35±0.16 | 18.44±0.57 | 31.09±0.61 | 7.5 |
| GraNd-score | 23.59±0.19 | 30.69±0.13 | 23.15±0.56 | 31.58±0.95 | 6.3 |
| EL2N-score | 24.60±0.81 | 31.49±0.33 | 26.62±0.34 | 33.91±0.56 | 4.3 |
| Optimization-based | 25.12±0.34 | 30.52±0.89 | 28.87±1.25 | 34.08±1.92 | 4.0 |
| Self-sup.-selection | 26.33±0.21 | 33.23±0.26 | 26.48±0.37 | 33.54±0.46 | 3.5 |
| Moderate-DS | **29.65±0.68** | **35.89±0.53** | **32.30±0.38** | **38.66±0.29** | **1.0** |
| Tiny-ImageNet with 20% mislabeled data | | | | | |
| Method / Selection ratio | 20% (V) | 30% (V) | 20% (S) | 30% (S) | Ave. rank ↓ |
| Random | 23.29±1.12 | 28.18±1.84 | 25.08±1.32 | 31.44±1.21 | 3.0 |
| Herding | **23.99±0.36** | 28.57±0.40 | 26.25±0.47 | 30.73±0.28 | 2.3 |
| Forgetting | 14.52±0.66 | 21.75±0.23 | 15.70±0.29 | 22.31±0.35 | 8.0 |
| GraNd-score | 22.44±0.46 | 27.95±0.29 | 23.64±0.10 | 30.85±0.21 | 4.3 |
| EL2N-score | 15.15±1.25 | 23.36±0.30 | 18.01±0.44 | 24.68±0.34 | 7.0 |
| Optimization-based | 22.93±0.58 | 24.92±2.50 | 25.82±1.70 | 30.19±0.48 | 4.5 |
| Self-sup.-selection | 18.39±1.30 | 25.77±0.87 | 22.87±0.54 | 29.80±0.36 | 5.8 |
| Moderate-DS | 23.68±0.19 | **28.93±0.19** | **28.82±0.33** | **32.39±0.21** | **1.3** |
| Tiny-ImageNet with adversarial examples by PGD attacks | | | | | |
| Method / Selection ratio | 20% (V) | 30% (V) | 20% (S) | 30% (S) | Ave. rank ↓ |
| Random | 26.12±1.09 | 31.98±0.78 | 28.28±0.90 | 34.59±1.18 | 3.3 |
| Herding | 26.76±0.59 | 32.56±0.35 | 28.87±0.48 | 35.43±0.22 | 2.0 |
| Forgetting | 24.55±0.57 | 31.83±0.36 | 23.32±0.37 | 31.82±0.15 | 6.0 |
| GraNd-score | 25.19±0.33 | 31.46±0.54 | 26.03±0.66 | 33.22±0.24 | 5.3 |
| EL2N-score | 21.73±0.47 | 27.66±0.32 | 22.66±0.35 | 29.89±0.64 | 8.0 |
| Optimization-based | 26.02±0.36 | 31.64±1.75 | 27.93±0.47 | 34.82±0.96 | 4.0 |
| Self-sup.-selection | 22.36±0.30 | 28.56±0.50 | 25.35±0.27 | 32.57±0.13 | 6.5 |
| Moderate-DS | **27.24±0.36** | **32.90±0.31** | **29.06±0.28** | **35.89±0.53** | **1.0** |

Table 8: Mean and standard deviation of experimental results (%) on Tiny-ImageNet. VGG-16 (V) and ShuffleNet (S) are exploited. The best result in each case is in bold.

| CIFAR-100 with ViT-small | | | | | |
|---|---|---|---|---|---|
| Method / Selection ratio | C | C+C | C+M | C+A | Ave. rank ↓ |
| Random | 33.80±0.54 | 31.29±0.61 | 26.67±0.54 | 31.01±0.45 | 3.3 |
| Herding | 32.16±0.37 | 31.75±0.22 | **32.27±0.53** | 31.28±0.66 | 2.5 |
| Forgetting | 33.52±0.73 | 24.45±0.29 | 26.24±1.07 | 28.26±1.95 | 5.3 |
| GraNd-score | 22.49±0.47 | 18.40±0.11 | 22.13±0.90 | 19.27±1.27 | 8.0 |
| EL2N-score | 26.15±0.21 | 23.27±0.68 | 24.37±0.82 | 22.06±1.68 | 7.0 |
| Optimization-based | 31.84±0.63 | 30.12±0.73 | 30.11±0.70 | 29.36±0.75 | 4.5 |
| Self-sup.-selection | 33.35±0.31 | 30.72±0.90 | 29.16±0.27 | 28.49±0.56 | 4.3 |
| Moderate-DS | **34.43±0.32** | **32.73±0.35** | 31.86±0.49 | **32.61±0.40** | **1.3** |

Table 9: Mean and standard deviation of experimental results (%) on simulated CIFAR-100 with transformer ViT-small. The selection ratio is 20% consistently. The best result in each case is in bold.

| CIFAR-100/Tiny-ImageNet with 35% mislabeled data | | | | |
|---|---|---|---|---|
| Method / Selection ratio | 20% (C) | 30% (C) | 20% (T) | 30% (T) | Ave. rank ↓ |
| Random | 24.51±1.34 | 32.26±0.81 | 14.64±0.29 | 19.41±0.45 | 4.8 |
| Herding | 29.42±1.54 | 37.50±2.12 | 15.14±0.45 | 20.19±0.45 | **2.0** |
| Forgetting | **29.48±1.98** | **38.01±2.21** | 11.25±0.90 | 17.07±0.66 | 3.8 |
| GraNd-score | 23.03±1.05 | 34.83±2.01 | 13.68±0.46 | 19.51±0.45 | 5.0 |
| EL2N-score | 21.95±1.08 | 31.63±2.84 | 10.11±0.25 | 13.69±0.32 | 8.0 |
| Optimization-based | 26.77±0.15 | 35.63±0.92 | 12.37±0.68 | 18.52±0.90 | 4.5 |
| Self-sup.-selection | 23.12±1.47 | 34.85±0.68 | 11.23±0.32 | 17.76±0.69 | 6.0 |
| Moderate-DS | 28.45±0.53 | 36.55±1.26 | **15.27±0.38** | **20.33±0.28** | **2.0** |

Table 10: Mean and standard deviation of experimental results (%) on CIFAR-100 (C) and Tiny-ImageNet (T) with 35% mislabeled data. The best result in each case is in bold.

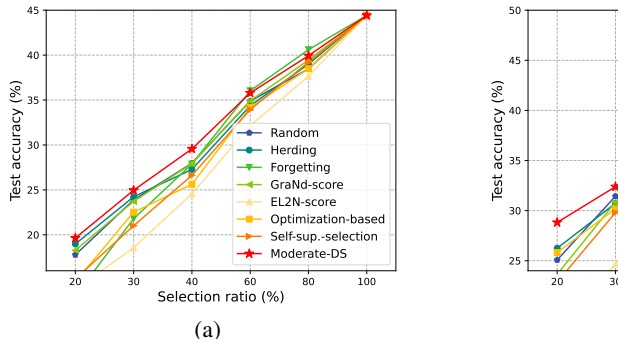
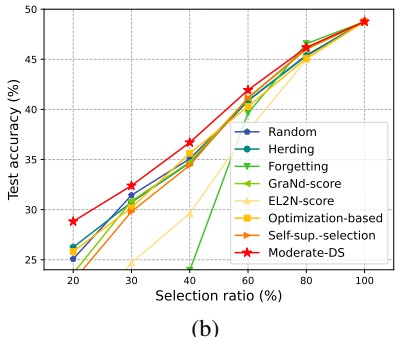

(a)                                         (b)

Figure 7: Illustrations of experimental results on Tiny-ImageNet with 20% mislabeled data.

| Tiny-ImageNet with 20% mislabeled data using ResNet-50 | | | | | | |
|---|---|---|---|---|---|---|
| Method / Selection ratio | 20% | 30% | 40% | 60% | 80% | 100% | Ave. rank ↓ |
| Random | 17.78±0.44 | 23.88±0.42 | 27.97±0.39 | 34.88±0.51 | 38.47±0.40 | 44.42±0.47 | 4.0 |
| Herding | 18.98±0.44 | 24.23±0.29 | 27.28±0.31 | 34.36±0.29 | 39.00±0.49 | 44.42±0.47 | 3.8 |
| Forgetting | 13.20±0.38 | 21.79±0.43 | 27.89±0.22 | **36.03±0.24** | **40.60±0.31** | 44.42±0.47 | 4.0 |
| GraNd-score | 18.28±0.32 | 23.72±0.18 | 27.93±0.33 | 34.91±0.19 | 39.45±0.45 | 44.42±0.47 | 3.2 |
| EL2N-score | 13.93±0.69 | 18.57±0.31 | 24.56±0.34 | 32.14±0.49 | 37.64±0.41 | 44.42±0.47 | 7.8 |
| Optimization-based | 14.77±0.95 | 22.52±0.77 | 25.62±0.90 | 34.18±0.79 | 38.49±0.69 | 44.42±0.47 | 6.0 |
| Self-sup.-selection | 15.10±0.73 | 21.01±0.36 | 26.62±0.22 | 33.93±0.36 | 39.22±0.12 | 44.42±0.47 | 5.8 |
| Moderate-DS | **19.64±0.40** | **24.96±0.30** | **29.56±0.21** | 35.79±0.36 | 39.93±0.23 | 44.42±0.47 | **1.4** |
| Tiny-ImageNet with 20% mislabeled data using ShuffleNet V2 | | | | | | |
| Method / Selection ratio | 20% | 30% | 40% | 60% | 80% | 100% | Ave. rank ↓ |
| Random | 25.08±1.32 | 31.44±1.21 | 35.07±0.69 | 40.88±1.00 | 45.40±0.84 | 48.76±0.27 | 3.6 |
| Herding | 26.25±0.47 | 30.73±0.28 | 34.67±0.36 | 40.86±0.45 | 45.34±0.38 | 48.76±0.27 | 4.4 |
| Forgetting | 15.70±0.29 | 22.31±0.35 | 23.94±0.31 | 39.64±0.45 | **46.55±0.43** | 48.76±0.27 | 6.4 |
| GraNd-score | 23.64±0.10 | 30.85±0.21 | 34.76±0.53 | 41.12±0.46 | 45.98±0.41 | 48.76±0.27 | 3.8 |
| EL2N-score | 18.01±0.44 | 24.68±0.34 | 29.63±0.46 | 37.87±0.36 | 44.97±0.48 | 48.76±0.27 | 7.4 |
| Optimization-based | 25.82±1.70 | 30.19±0.48 | 35.62±0.77 | 40.29±0.56 | 45.11±0.41 | 48.76±0.27 | 4.6 |
| Self-sup.-selection | 22.87±0.54 | 29.80±0.36 | 34.42±0.68 | 41.16±0.34 | 46.05±0.09 | 48.76±0.27 | 4.6 |
| Moderate-DS | **28.82±0.33** | **32.39±0.21** | **36.70±0.44** | **41.93±0.25** | 46.19±0.52 | 48.76±0.27 | **1.2** |

Table 11: Mean and standard deviation of experimental results (%) on Tiny-ImageNet with 20% mislabeled data. ResNet-50 and ShuffleNet V2 are exploited. The average rank is calculated with the ranks of a method in 20%, 30%, 40%, 60%, and 80% cases. The best result in each case is in bold.

# E  TECHNICAL DETAILS OF THE MUTUAL INFORMATION ESTIMATOR

In the main paper, we employ the mutual information estimator (MINE) (Belghazi et al., 2018) to justify our claims. Here, we discuss the technical details of the estimator. Given two random variables $X$ and $Z$, and $n$ i.i.d. examples drawn from a joint distribution $\mathbb{P}_{XZ}$, MINE is defined as

$$\widehat{I(X;Z)}_n = \sup_{\theta \in \Theta} \mathbb{E}_{\mathbb{P}_{XZ}^{(n)}}[T_\theta] - \log(\mathbb{E}_{\mathbb{P}_{XZ}^{(n)} \otimes \hat{\mathbb{P}}_X^{(n)}}[e^{T_\theta}]), \quad (3)$$

| CIFAR-100/Tiny-ImageNet with 30% corrupted images | | | | |
|---|---|---|---|---|
| Method / Selection ratio | 20% (C) | 30% (C) | 20% (T) | 30% (T) | Ave. rank ↓ |
| Random | 39.67±1.72 | 48.89±1.68 | 17.34±1.28 | 23.14±0.96 | 4.3 |
| Herding | **43.25±1.11** | **53.24±1.06** | 12.04±0.66 | 18.03±0.58 | 4.5 |
| Forgetting | 26.59±1.29 | 39.67±1.95 | 13.68±0.45 | 21.10±0.32 | 6.8 |
| GraNd-score | 31.96±1.58 | 45.19±1.61 | 14.85±0.39 | 21.06±0.35 | 6.0 |
| EL2N-score | 20.53±0.89 | 26.62±1.13 | 16.60±0.48 | 23.90±0.15 | 6.3 |
| Optimization-based | 33.14±0.62 | 40.86±2.30 | 18.55±0.16 | 25.73±0.82 | 4.3 |
| Self-sup.-selection | 41.53±0.83 | 52.52±1.38 | 20.09±0.40 | 26.07±0.26 | 2.5 |
| Moderate-DS | 42.58±0.64 | 53.14±0.89 | **20.65±0.12** | **27.32±0.36** | **1.5** |

Table 12: Mean and standard deviation of experimental results (%) on CIFAR-100 (C) and Tiny-ImageNet (T) with 30% corrupted images. The best result in each case is in bold.

| Tiny-ImageNet with simultaneous corrupted, mislabeled, and adversarial examples | | | |
|---|---|---|---|
| Method / Selection ratio | 20% | 30% | 40% | Ave. rank ↓ |
| Random | 19.08±0.56 | 24.69±0.59 | 28.72±0.89 | 3.7 |
| Herding | 19.08±0.23 | 24.63±0.19 | 28.65±0.16 | 4.0 |
| Forgetting | 15.96±0.21 | 23.79±0.17 | 29.23±0.19 | 5.0 |
| GraNd-score | 18.91±0.34 | 25.22±0.15 | 28.57±0.45 | 4.3 |
| EL2N-score | 15.36±0.25 | 21.18±0.21 | 25.83±0.52 | 8.0 |
| Optimization-based | 19.05±1.00 | 24.93±0.46 | 29.09±1.02 | 3.3 |
| Self-sup.-selection | 17.46±0.60 | 23.79±0.29 | 28.45±0.28 | 6.7 |
| Moderate-DS | **20.91±0.36** | **26.24±0.19** | **30.31±0.10** | **1.0** |

Table 13: Mean and standard deviation of experimental results (%) Tiny-ImageNet with simultaneous corrupted, mislabeled, and adversarial examples. The best result in each case is in bold.

where $T_\theta$ is parameterized by a deep neural network with parameters $\theta$, and $\hat{\mathbb{P}}^n$ is the empirical distribution associated to $n$ i.i.d. examples. After the definition, we follow (Belghazi et al., 2018) to give the procedure of MINE for the estimation of mutual information (see Algorithm 1).

---

**Algorithm 1** MINE

$\theta \leftarrow$ initialize network parameters
**repeat**
    Draw $b$ minibatch examples from the joint distribution: $(\boldsymbol{x}^{(1)}, \boldsymbol{z}^{(1)}), \ldots, (\boldsymbol{x}^{(b)}, \boldsymbol{z}^{(b)}) \sim \mathbb{P}_{XZ}$
    Draw $b$ examples from the $Z$ marginal distribution: $\bar{\boldsymbol{z}}^{(1)}, \ldots, \bar{\boldsymbol{z}}^{(b)} \sim \mathbb{P}_Z$
    Evaluate the lower-bound: $\mathcal{V}(\theta) \leftarrow \frac{1}{b} \sum_{i=1}^{b} T_\theta(\boldsymbol{x}^{(i)}, \boldsymbol{z}^{(i)}) - \log(\frac{1}{b} \sum_{i=1}^{b} e^{T_\theta(\boldsymbol{x}^{(i)}, \bar{\boldsymbol{z}}^{(i)})})$
    Evaluate bias corrected gradients (e.g., moving average): $\widehat{G}(\theta) \leftarrow \widetilde{\nabla}_\theta \mathcal{V}(\theta)$
    Update the statistics network parameters: $\theta \leftarrow \theta + \widehat{G}(\theta)$
**until** convergence

---

