# OpenReview forum: "Moderate Coreset: A Universal Method of Data Selection for Real-world Data-efficient Deep Learning"
_ICLR.cc/2023/Conference — ICLR 2023 poster_

### Official Review · Reviewer_Ergu · 2022-10-18

**Confidence:** 3
**Correctness:** 3
**Technical Novelty And Significance:** 3
**Empirical Novelty And Significance:** 3
**Recommendation:** 8

**Clarity, Quality, Novelty And Reproducibility:**

Clarity:
* Although the idea is easy to follow and to implement, I suggest the authors to encapsulate the idea into one algorithm environment. The intuition behind the idea clearly stated.

Quality:
* The solution is very well presented. The experimental results are quite promising. However, I would like to see how the algorithm behaves when different architectures are provided. The algorithm was only tested using a resnet solution.

Novelty:
* Although the idea is somehow simple, its novelty is clear.

Reproducibility:
* The algorithm is easy to understand and to reproduce. The model hyper-parameters are included in the paper.

**Strength And Weaknesses:**

Strength:
* The algorithm is easy to understand and to implement
* Both computational and spatial costs are minimal
* The experimental results are promising

Weaknesses
* Only tested in one neural network architecture

**Summary Of The Paper:**

The authors propose a novel method for reducing the dataset size by selecting the data that is close to the median error of the projection of each sample against the mean projection of all samples that belongs to the same class. The authors evaluate their results in three datasets, obtaining state-of-the-art results in almost all of them.

**Summary Of The Review:**

Overall, I think it is a very interesting idea. It is very simple, efficient and easy to implement. I am lowering my score because of the lack of multiple NN architectures in the experimental section. As a future remark, I would like to see if a combination between this approach and the forgetting algorithm could combine the benefits of both methods.

---

> ### Author Response · Authors · 2022-11-17
> **Response to Reviewer Ergu (1/2)**
>
> Thanks for your comments. We address your concerns as follows.
>
> > **Q1**: Experiments with multiple network architectures.
>
> **A1**: Thanks for your suggestion. We employ more network architectures to verify the effectiveness of our method, i.e., VGG-16  (abbreviated as V) [d1] and ShuffleNetv2  (abbreviated as S) [d2], which are popularly used in the research community. We conduct experiments on Tiny-ImageNet. As did in the main paper, we consider more challenging cases of small selection ratios (i.e., 20% and 30%) and employ the following settings:
> - Experiments with original datasets (see Table d-1);
> - Experiments with 20% corrupted images (see Table d-2);
> - Experiments with 20% mislabeled data (see Table d-3);
> - Experiments with adversarial examples by PGD attacks (see Table d-4).
> As can be seen, with different network architectures, our method consistently works well. These experiments are also added to Appendix C.1 of the revised paper.
>
> | Methods     | 20% (V) | 30% (V) | 20% (S) | 30% (S) | Ave. rank |
> | :---------- | :-------: | :-------: | :-------: | :-------: | :-------: |
> | Random      | 29.63$\pm$0.43 | 35.38$\pm$0.83 | 32.40$\pm$1.06 | 39.13$\pm$0.81 | 3.5 |
> | Herding     | 31.05$\pm$0.22 | 36.27$\pm$0.57 | 33.10$\pm$0.39 | 38.65$\pm$0.22 | 2.3 |
> | Forgetting  | 27.53$\pm$0.36 | 35.61$\pm$0.39 | 27.82$\pm$0.56 | 36.26$\pm$0.51 | 5.8 |
> | GraNd-score | 29.93$\pm$0.95 | 35.61$\pm$0.39 | 29.56$\pm$0.46 | 37.40$\pm$0.38 | 4.3 |
> | EL2N-score |26.47$\pm$0.31|33.19$\pm$0.51|28.18$\pm$0.27|35.81$\pm$0.29|7.3|
> | Optimization-based |25.92$\pm$0.64|34.82$\pm$1.29|31.37$\pm$1.14|38.22$\pm$0.78|5.3|
> | Self-sup.-selection |25.16$\pm$1.10|33.30$\pm$0.94|29.47$\pm$0.56|36.68$\pm$0.36|6.8|
> |Moderate-DS|**31.45$\pm$0.32**|**37.89$\pm$0.36**|**33.32$\pm$0.41**|**39.68$\pm$0.34**|**1.0**|
>
> Table d-1: Experiments with original datasets under different network architectures.
>
> | Methods             |      20% (V)       |      30% (V)       |      20% (S)       |      30% (S)       | Ave. rank |
> | :------------------ | :----------------: | :----------------: | :----------------: | :----------------: | :-------: |
> | Random              |   26.33$\pm$0.88   |   31.57$\pm$1.31   |   29.15$\pm$0.83   |   34.72$\pm$1.00   |    2.5    |
> | Herding             |   18.03$\pm$0.33   |   25.77$\pm$0.34   |   23.33$\pm$0.43   |   31.73$\pm$0.38   |    7.0    |
> | Forgetting          |   19.41$\pm$0.57   |   28.35$\pm$0.16   |   18.44$\pm$0.57   |   31.09$\pm$0.61   |    7.5    |
> | GraNd-score         |   23.59$\pm$0.19   |   30.69$\pm$0.13   |   23.15$\pm$0.56   |   31.58$\pm$0.95   |    6.3    |
> | EL2N-score          |   24.60$\pm$0.81   |   31.49$\pm$0.33   |   26.62$\pm$0.34   |   33.91$\pm$0.56   |    4.3    |
> | Optimization-based  |   25.12$\pm$0.34   |   30.52$\pm$0.89   |   28.87$\pm$1.25   |   34.08$\pm$1.92   |    4.0    |
> | Self-sup.-selection |   26.33$\pm$0.21   |   33.23$\pm$0.26   |   26.48$\pm$0.37   |   33.54$\pm$0.46   |    3.5    |
> | Moderate-DS         | **29.65$\pm$0.68** | **35.89$\pm$0.53** | **32.30$\pm$0.38** | **38.66$\pm$0.29** |  **1.0**  |
>
> Table d-2: Experiments with 20% corrupted images under different network architectures.
>
> | Methods             |      20% (V)       |      30% (V)       |      20% (S)       |      30% (S)       | Ave. rank |
> | :------------------ | :----------------: | :----------------: | :----------------: | :----------------: | :-------: |
> | Random              |   23.29$\pm$1.12   |   28.18$\pm$1.84   |   25.08$\pm$1.32   |   31.44$\pm$1.21   |    3.0    |
> | Herding             | **23.99$\pm$0.36** |   28.57$\pm$0.40   |   26.25$\pm$0.47   |   30.73$\pm$0.28   |    2.3    |
> | Forgetting          |   14.52$\pm$0.66   |   21.75$\pm$0.23   |   15.70$\pm$0.29   |   22.31$\pm$0.35   |    8.0    |
> | GraNd-score         |   22.44$\pm$0.46   |   27.95$\pm$0.29   |   23.64$\pm$0.10   |   30.85$\pm$0.21   |    4.3    |
> | EL2N-score          |   15.15$\pm$1.25   |   23.36$\pm$0.30   |   18.01$\pm$0.44   |   24.68$\pm$0.34   |    7.0    |
> | Optimization-based  |   22.93$\pm$0.58   |  24.92$\pm$2.501   |   25.82$\pm$1.70   |   30.19$\pm$0.48   |    4.5    |
> | Self-sup.-selection |   18.39$\pm$1.30   |   25.77$\pm$0.87   |   22.87$\pm$0.54   |   29.80$\pm$0.36   |    5.8    |
> | Moderate-DS         |   23.68$\pm$0.19   | **28.93$\pm$0.19** | **28.82$\pm$0.33** | **32.39$\pm$0.21** |  **1.3**  |
>
> Table d-3: Experiments with 20% mislabeled data under different network architectures.

---

> > ### Author Response · Authors · 2022-11-17
> > **Response to Reviewer Ergu (2/2)**
> >
> > | Methods             |      20% (V)       |      30% (V)       |      20% (S)       |      30% (S)       | Ave. rank |
> > | :------------------ | :----------------: | :----------------: | :----------------: | :----------------: | :-------: |
> > | Random              |   26.12$\pm$1.09   |   31.98$\pm$0.78   |   28.28$\pm$0.90   |   34.59$\pm$1.18   |    3.3    |
> > | Herding             |   26.76$\pm$0.59   |   32.56$\pm$0.35   |   28.87$\pm$0.48   |   35.43$\pm$0.22   |    2.0    |
> > | Forgetting          |   24.55$\pm$0.57   |   31.83$\pm$0.36   |   23.32$\pm$0.37   |   31.82$\pm$0.15   |    6.0    |
> > | GraNd-score         |   25.19$\pm$0.33   |   31.46$\pm$0.54   |   26.03$\pm$0.66   |   33.22$\pm$0.24   |    5.3    |
> > | EL2N-score          |   21.73$\pm$0.47   |   27.66$\pm$0.32   |   22.66$\pm$0.35   |   29.89$\pm$0.64   |    8.0    |
> > | Optimization-based  |   26.02$\pm$0.36   |   31.64$\pm$1.75   |   27.93$\pm$0.47   |   34.82$\pm$0.96   |    4.0    |
> > | Self-sup.-selection |   22.36$\pm$0.30   |   28.56$\pm$0.50   |   25.35$\pm$0.27   |   32.57$\pm$0.13   |    6.5    |
> > | Moderate-DS         | **27.24$\pm$0.36** | **32.90$\pm$0.31** | **29.06$\pm$0.28** | **35.89$\pm$0.53** |  **1.0**  |
> >
> > Table d-4: Experiments with adversarial examples under different network architectures.
> >
> > >**Q2**: If a combination between this approach and the forgetting algorithm could combine the benefits of both methods.
> >
> > **A2**:  Thanks for your comments. We conduct experiments on simulated CIFAR-100. The selection ratios are set to 80%, where Forgetting achieves nice performance. Results in Table d-5 show that the moderate coreset and Forgetting can combine the benefits of both methods.
> >
> >
> >
> > | Methods     |  80% (CIFAR-100)   |
> > | :---------- | :----------------: |
> > | Forgetting  |   73.67$\pm$1.12   |
> > | Moderate-DS |   73.67$\pm$0.30   |
> > | Forgetting+ | **73.85$\pm$0.21** |
> >
> > Table d-5: Experiments of comparing Forgetting+ to Moderate-DS and Forgetting with 20% corrupted images.
> >
> > ----
> >
> > [d1] Karen Simonyan and Andrew Zisserman. Very Deep Convolutional Networks for Large-Scale Image Recognition. arXiv preprint:1409.1556.
> > [d2] Ningning Ma et al. Shufflenet v2: Practical Guidelines for Efficient Cnn Architecture Design. ECCV 2018.

---

> > > ### Comment · Reviewer_Ergu · 2022-11-27
> > > **Thank you for your response**
> > >
> > > I would like to thank the authors for their response. I am glad with the new additions, I think it is a good paper.

---

> ### Author Response · Authors · 2022-11-22
> **Further Discussion**
>
> Dear Reviewer Ergu,
>
> Thanks for your efforts in reviewing this paper and your constructive comments. We have provided responses to address your concerns, in particular experiments with multiple network architectures. Are there unclear explanations here? We could further clarify them.
>
> Best,
> Authors

---

> ### Author Response · Authors · 2022-11-26
> **Further Discussion**
>
> Dear Reviewer Ergu,
>
> Thanks for your efforts in reviewing this paper. Your suggestions improve this paper clearly. We have added experiments with various network architectures. Are there unclear explanations here? We could further clarify them.
>
> Thank you very much!
>
> Best,
> Authors

---

### Official Review · Reviewer_vgUK · 2022-10-24

**Confidence:** 4
**Correctness:** 3
**Technical Novelty And Significance:** 1
**Empirical Novelty And Significance:** 2
**Recommendation:** 5

**Clarity, Quality, Novelty And Reproducibility:**

Generally, the paper seems clear.

The results are not surprising, but that may be OK. I think the novelty over Herding  (and possibly random) needs to be better argued and supported. Downsides need to be better presented.

**Strength And Weaknesses:**

Pros:
-- The empirical results seem consistent with the intuition presented.
-- Overall, the paper is intuitive.

Cons:
-- Isn't Herding also a form of moderate corset strategy? Assuming that is the case, the paper could do a better job explaining the distinction and why that matters.
-- The paper claims that computing distances on hidden representation is more efficient. Any data to substantiate that? Any timing numbers?
 -- What is the intuition, if any, for outperforming Herding?  It may be good to discuss that further.
 -- Could you elaborate more on how you trained the model when corruptness  or mislabeling was injected in your experiments? For instance, when you had 20% mislabeled examples, what are the curves that are similar to Figure 5? I didn't find them in the Appendix.
Did the models overfit to the corrupted or mislabeled data?








**Summary Of The Paper:**

The paper argues that corset should be selected  from near the score median. The authors argue that corset examples that are closer to the median are expected to generalize better to different scenarios than those  that pick examples based on highest loss. The authors point out that in prior work, where outliers are expected to be significant, data points with smaller losses are favored. Their approach seems similar but instead of picking examples based on loss, it picks examples that are closer to the mean class-specific center in feature space, where closeness is measured as as ||z  - zj ||_2, where zj is the class center.

The abstract and intro does not make it apparent what the scenarios that authors have in mind. When reading the paper, I was thinking they  are going to present examples of distribution shifts (e.g., different weightage of unrepresented categories). But, later, in the empirical evaluations, the focus was on data corruption -- mislabeled examples or corrupted images (e.g., fog, motion blur, random occlusion). Unsurprisingly, these are also the scenarios where one can expect techniques that prefer high loss and those close to decision boundary to underperform So, I am not entirely surprised that their scheme did well with increased data corruption or mislabeling compared to most schemes. But, perhaps unsurprisingly, they underperform in the absence of corruption and mislabeling, especially if the corset is large.

The authors tested on adversarial inputs that attack an original classification pipeline, but one weakness in that testing is that the adversarial inputs do not seem adaptive to their technique.







**Summary Of The Review:**

The paper makes a cogent argument as to why a coreset that has examples from near the center of the each class may be more robust to outliers or mislabeled data. But, Herding  (Welling 2009) also does something similar. So, that by itself does not make the paper novel. The authors could do a better job articulating why Herding is not the solution they are looking for. If the main argument is the proposed scheme is more performance-efficient in terms of selecting a corset on very large datasets, then the entire set of experiments should have been geared to supporting that argument.  So, overall, I think interesting ideas are possibly there in the paper. But, they need to be better teased out and better supported.

---

> ### Author Response · Authors · 2022-11-17
> **Response to Reviewer vgUK (1/3)**
>
> Thanks for your comments. We address your concerns as follows.
>
> >**Q1**: The abstract and intro do not make it apparent what the scenarios that authors have in mind.
>
> **A1**: Thanks for your comment. We address your concerns as follows.
> - The scenarios of corrupted images and mislabeled data are representative of distribution shifts. Specifically, in supervised learning, training data are sampled from the joint distribution $p(X, Y)=p(X)p(Y|X)$. We tend to learn the distribution for generalization. For the case of corrupted images, the joint distribution is shifted due to the change of $p(X)$. For the case of mislabeled data, the joint distribution is shifted due to the change of $p(Y|X)$.
> - We do not mention distribution shifts explicitly, as the paper does not only target distribution-shift scenarios. For example, in Section 2.2, we also discuss the scenarios, where the demands of coreset sizes are always changed or access to network architectures is changed.
> - We agree with you that baselines preferring data points with high losses and close to decision boundary can perform well when there is no corruption and mislabeling. However, the proposed method still outperforms them in lots of experimental cases (see Tables 5 and 6 of Appendix A).
>
> >**Q2**: The authors tested on adversarial inputs that attack an original classification pipeline, but one weakness in that testing is that the adversarial inputs do not seem adaptive to their technique.
>
> **A2**: We address your question as follows.
> - We consider the setting where adversarial noise has been injected into original datasets. Then, we verify the robustness of a method against adversarial noise. The setting is commonly used in previous works on adversarial learning, such as [c1,c2]. The setting and achieved results also verify the effectiveness of the proposed method.
> - We do not exploit adaptive adversarial attacks, since it is hard to ensure the subsequent comparison is fair. Specifically, we have multiple data selection methods. If adaptive adversarial attacks are introduced, for comparison, we need to design multiple adaptive adversarial attacks for these methods. This is beyond the scope of this paper. Besides, since adversarial attacks are different, the subsequent comparison of the data-selection performance is not fair.
>
> >**Q3**:  Isn't Herding also a form of moderate coreset strategy?
>
> **A3**: **No, Herding is not a form of moderate coreset strategy**. We emphasize that
> - It is not equivalent to the moderate-coreset concept and data selection by using extracted representations.
> - Based on any score criterion, the moderate coreset is built by data points with scores that are close to the score median. The moderate coreset is a concept, which is not limited to a specific method.
> -  If the score criterion is the distance to class centers, Herding selects data points with smaller scores, but not scores close to the score median. Therefore, Herding is not related to the moderate coreset. It is different from our Moderate-DS.
>
> >**Q4**: The paper claims that computing distances on hidden representation are more efficient. Any timing numbers?
>
> **A4**: Thanks. The reason for this claim is that, compared with multiple methods, the proposed method only needs computing distances on hidden representations without model retraining and other complex operations. Therefore, clearly, it is more efficient. The explanations are also provided in Section 4.1. Below, in Table c-1, we provide a comparison of the used time for data selection. The experimental settings are kept the same in the main paper.
>
>
>
> | Methods             | On CIFAR-100 | On Tiny-ImageNet |
> | :------------------ | :----------: | :--------------: |
> | Random              |      -       |        -         |
> | Herding             |    ~1min     |      ~1min       |
> | Forgetting          |   ~45mins    |     ~158mins     |
> | GraNd-score         |   ~50mins    |     ~106mins     |
> | EL2N-score          |   ~15mins    |     ~12mins      |
> | Optimization-based  |   ~453mins   |     ~815mins     |
> | Self-sup.-selection |   ~135mins   |     ~202mins     |
> | Moderate-DS         |    ~1min     |      ~1min       |
>
> Table c-1: Time consumption comparison between different data selection methods.

---

> > ### Author Response · Authors · 2022-11-17
> > **Response to Reviewer vgUK (2/3)**
> >
> > >**Q5**: The intuition for outperforming Herding?
> >
> > **A5**: Thanks. Note that in A3, we have explained that Herding is different from our methods and is not a moderate coreset strategy. Here, we further address your concerns. Based on the score about the distance to class centers, Herding selects examples with smaller scores, i.e., easy examples. Although the easy examples are more robust against outliers, the covariate-shift problem [c3,c4] will occur if we only employ them during training, since the distribution of selected examples will be much different from the distribution of full examples. The problem will degrade performance. Compared with Herding, our Moderate-DS selects examples with scores that are close to the score median, it can achieve a great trade-off between robustness against outliers and against covariate shift, leading to better performance.
> >
> > > **Q6**: Several concerns on experiments with mislabeled data.
> >
> > **A6**: We address your concerns on experiments with mislabeled data as follows.
> >
> > - When corruptness or mislabeling was injected in experiments, we train models with the same configures as the experiments without corruptness or mislabeling, which are described in Section 4.1.
> >
> > - Comparing the results of experiments with/without corruptness (Tables 5 and 6 in Appendix A), the accuracy drops can explain that the models overfit to the corrupted data.
> > - In the main paper, we consider more challenging cases, where the selection ratio is small (20% and 30%). Therefore, the curve of "Test accuracy vs. Selection ratio" is not provided. Here, we provide experiments with 40%, 60%, 80%, and 100%, when the label noise rate is 20%. Due to the limited rebuttal time, we conduct experiments on Tiny-ImageNet. The results are shown in Table c-2 and also provided in Appendix C.3 of the revised paper.
> >
> > | Methods             | 20%  | 30%  | 40%  | 60%  | 80% | 100% | Ave. rank |
> > | :------------------ | :--: | :--: | :--: | :--: | :--: | :--: | :--: |
> > | Random              | 17.78$\pm$0.44 | 23.88$\pm$0.42 | 27.97$\pm$0.39 | 34.88$\pm$0.51 | 38.47$\pm$0.40 |44.42$\pm$0.47|4.0|
> > | Herding             | 18.98$\pm$0.44 | 24.23$\pm$0.29 | 27.28$\pm$0.31 | 34.36$\pm$0.29 | 39.00$\pm$0.49 |44.42$\pm$0.47|3.8|
> > | Forgetting          | 13.20$\pm$0.38 | 21.79$\pm$0.43 | 27.89$\pm$0.22 | **36.03$\pm$0.24** | **40.60$\pm$0.31** |44.42$\pm$0.47|4.0|
> > | GraNd-score         | 18.28$\pm$0.32 | 23.72$\pm$0.18 | 27.93$\pm$0.33 | 34.91$\pm$0.19 | 39.45$\pm$0.45 |44.42$\pm$0.47|3.2|
> > | EL2N-score          | 13.93$\pm$0.69 | 18.57$\pm$0.31 | 24.56$\pm$0.34 | 32.14$\pm$0.49 | 37.64$\pm$0.41 |44.42$\pm$0.47|7.8|
> > | Optimization-based  | 14.77$\pm$0.95 | 22.52$\pm$0.77 | 25.62$\pm$0.90 | 34.18$\pm$0.79 | 38.49$\pm$0.69 |44.42$\pm$0.47|6.0|
> > | Self-sup.-selection | 15.10$\pm$0.73 | 21.01$\pm$0.36 | 26.62$\pm$0.22 | 33.93$\pm$0.36 | 39.22$\pm$0.12 |44.42$\pm$0.47|5.8|
> > | Moderate-DS         | **19.64$\pm$0.40** | **24.96$\pm$0.30** | **29.56$\pm$0.21** | 35.79$\pm$0.36 | 39.93$\pm$0.23 |44.42$\pm$0.47|**1.4**|
> >
> > Table c-2: Experimental results on Tiny-ImageNet with mislabeled data. ResNet-50 is employed.
> >
> > | Methods             |        20%         |        30%         |        40%         |        60%         |        80%         |      100%      | Ave. rank |
> > | :------------------ | :----------------: | :----------------: | :----------------: | :----------------: | :----------------: | :------------: | :------------: |
> > | Random              |   25.08$\pm$1.32   |   31.44$\pm$1.21   |   35.07$\pm$0.69   |   40.88$\pm$1.00   |   45.40$\pm$0.84   | 48.76$\pm$0.27 |3.6|
> > | Herding             |   26.25$\pm$0.47   |   30.73$\pm$0.28   |   34.67$\pm$0.36   |   40.86$\pm$0.45   |   45.34$\pm$0.38   | 48.76$\pm$0.27 |4.4|
> > | Forgetting          |   15.70$\pm$0.29   |   22.31$\pm$0.35   |   23.94$\pm$0.31   |   39.64$\pm$0.45   | **46.55$\pm$0.43** | 48.76$\pm$0.27 |6.4|
> > | GraNd-score         |   23.64$\pm$0.10   |   30.85$\pm$0.21   |   34.76$\pm$0.53   |   41.12$\pm$0.46   |   45.98$\pm$0.41   | 48.76$\pm$0.27 |3.8|
> > | EL2N-score          |   18.01$\pm$0.44   |   24.68$\pm$0.34   |   29.63$\pm$0.46   |   37.87$\pm$0.36   |   44.97$\pm$0.48   | 48.76$\pm$0.27 |7.4|
> > | Optimization-based  |   25.82$\pm$1.70   |   30.19$\pm$0.48   |   35.62$\pm$0.77   |   40.29$\pm$0.56   |   45.11$\pm$0.41   | 48.76$\pm$0.27 |4.6|
> > | Self-sup.-selection |   22.87$\pm$0.54   |   29.80$\pm$0.36   |   34.42$\pm$0.68   |   41.16$\pm$0.34   |   46.05$\pm$0.09   | 48.76$\pm$0.27 |4.6|
> > | Moderate-DS         | **28.82$\pm$0.33** | **32.39$\pm$0.21** | **36.70$\pm$0.44** | **41.93$\pm$0.25** |   46.19$\pm$0.52   | 48.76$\pm$0.27 |**1.2**|
> >
> > Table c-3: Experimental results on Tiny-ImageNet with mislabeled data. ShuffleNet V2 is employed.

---

> > > ### Author Response · Authors · 2022-11-17
> > > **Response to Reviewer vgUK (3/3)**
> > >
> > > >**Q7**: The paper makes a cogent argument as to why a coreset that has examples from near the center of each class may be more robust to outliers or mislabeled data. But, Herding also does something similar. So, that by itself does not make the paper novel.
> > >
> > > **A7**: We agree with you that Herding makes the argument as to why a coreset that has examples from near the center of each class may be more robust to outliers or mislabeled data. However, the argument is not the main claim of this paper and not the contribution of this paper.
> > >
> > > This paper claims that, with any score criterion, the moderate coreset is built by data points with scores that are close to the score median. The moderate coreset is a concept. The method Moderate-DS is a proof-of-concept. As a distribution can be generally depicted by the median of scores, data points with scores that are close to the score median can be seen as a proxy of all data points, which generalize different scenarios. Learning with outliers or mislabeled data is only one experimental setting, where Herding also works well. We also exploit diverse settings to support our claims, as discussed in the main paper.
> > >
> > > As the concern mainly sources from the method Herding, we stress
> > >
> > > - The difference between Herding and our Moderate-DS (check **A3**);
> > > -  Why Herding is not related to the proposed moderate coreset of this paper (check **A3**);
> > > -  The explanation of why our method outperforms Herding (check **A5**).
> > >
> > > >**Q8**: On the novelty and contributions of this paper.
> > >
> > > **A8**: We believe this paper is novel and beneficial. The contributions are sufficient. We have summarized contributions in the main paper, which are also appreciated by the other three reviewers.
> > >
> > >
> > >
> > > ----
> > >
> > > [c1] Hongyang Zhang et al. Theoretically Principled Trade-off between Robustness and Accuracy. ICML 2019.
> > > [c2] Yisen Wang et al. Improving Adversarial Robustness Requires Revisiting Misclassified Examples. ICLR 2019.
> > > [c3] Masashi Sugiyama and Motoaki Kawanabe. Machine Learning in Non-Stationary Environments: Introduction to Covariate Shift Adaptation. MIT Press 2012.
> > > [c4] Antonin Berthon et al. Confidence Scores Make Instance-dependent Label-noise Learning Possible. ICML 2021.

---

> ### Author Response · Authors · 2022-11-20
> **Further Discussions**
>
> Dear Reviewer vgUK,
>
> Thanks a lot for your efforts in reviewing this paper. We tried our best to address the mentioned concerns. Are there unclear explanations here? We could further clarify them.
>
> Best,
> Authors

---

> ### Author Response · Authors · 2022-11-22
> **Further Discussions**
>
> Dear Reviewer vgUK,
>
> Thanks again for your efforts in reviewing. We are still looking forward to your reply. Would you mind checking our response and confirming if there are unclear explanations?
>
> Best,
> Authors

---

> > ### Comment · Reviewer_vgUK · 2022-11-22
> > **Followup clarifications**
> >
> > Thanks for your response. Some clarification questions below:
> >
> > In Herding vs. your method, can you elaborate a bit more? Is the conceptual difference  use of center (or mean) in Herding versus use of median in your method? But, both try to pick examples from near the "center", but "center" is defined differently?
> >
> > Did you do experiments on CIFAR-10 as well? It seems that has been used in most of the earlier papers and seems like a natural benchmark to evaluate on.  If so, what were the results?
> >
> > Is CIFAR-100 used in previous papers on coresets with similar settings you used?  If so, could you point me to them?
> >
> > Also, what  is the command line to train your CIFAR-100 model and reproduce your results at different selection rates?
> >
> > Thanks.

---

> > > ### Author Response · Authors · 2022-11-22
> > > **Response**
> > >
> > > Thanks for your comments. We address your concerns as follows.
> > >
> > > > **Q1**: Herding vs. Our method.
> > >
> > > **A1**: To better explain the difference between Herding and our method, we provide the procedures of Herding and our method below.
> > >
> > > (1) The procedure of Herding
> > > - Obtain class centers with deep representations. The class center of a class is achieved with the mean of the representations from one class (Eq. (1) in the main paper).
> > > - Calculate the distances between every data point and its corresponding class center.
> > > - **Select the data points with smaller distances.**
> > >
> > > (2) The procedure of our method (Moderate-DS)
> > > - Obtain class centers with deep representations. The class center of a class is achieved with the mean of the representations from one class (Eq. (1) in the main paper).
> > > - Calculate the distances between every data point and its corresponding class center.
> > > - **Select the data points that are equipped with distances close to the median of distances. Notice that the median works with calculated distances. It is not equivalent to the median of the representations.**
> > >
> > > Therefore, only Herding picks examples that are close to class centers. Our method is different from Herding as mentioned. We suggest that you can check **Figure 1** in the main paper, where subfigure (a) represents Herding but subfigure (c) represents our method.
> > >
> > > > **Q2**: Experiments on CIFAR-10.
> > >
> > > **A2**: We do not conduct experiments on CIFAR-10 in this paper. CIFAR-100, Tiny-ImageNet, and ImageNet are popular benchmarks, which are argued to be more challenging than CIFAR-10. In the rebuttal, we also supplement lots of experiments to enhance this paper. We believe that the present experiments can demonstrate the effectiveness of our method. Due to the limited rebuttal time, we cannot supplement experiments on CIFAR-10 under all settings (i.e., permutations and combinations of different real-world cases, different selection ratios, different networks, etc). We appreciate your advice and leave this part for future work.
> > >
> > >
> > > >**Q3**: Concerns on CIFAR-100.
> > >
> > > **A3**: To our best knowledge, we are the first to design a universal algorithm of coreset construction for real-world data-efficient deep learning, where multiple realistic settings are considered. Prior works such as [1] and [2] also exploit CIFAR-100. However, they are limited to evaluations in ideal scenes.
> > >
> > > [1] Mariya Toneva et al. An empirical study of example forgetting during deep neural network learning. ICLR, 2019.
> > > [2] Mansheej Paul et al. Deep learning on a data diet: finding important examples early in training. NeurIPS, 2021.
> > >
> > >
> > > >**Q4**: What is the command line to train your CIFAR-100 model and reproduce your results at different selection rates?
> > >
> > > **A4**: We provide codes in the supplementary material. To reproduce the results at different selection rates, you can
> > > - Download the original CIFAR-100 dataset and put it into the file 'data'. It will be downloaded automatically if you run codes for the first time.
> > > - Run "python train-resnet-cifar100.py --save='True' --model_dir='save_file'". If there are pre-trained backbones provided, e.g., models in PyTorch Zoos, you can skip this step.
> > > - Revise Line 39 of "dump_features.py" to the path to the backbone and run "dump_features.py" to obtain deep representations of data points.
> > > - Run "cluster.py" to obtain distances between each data point and its class center.
> > > - Run "prune.py" to obtain the index of data points that will be pruned at different selection rates. Then, you can evaluate the quality of the obtained coreset.
> > >
> > > In order not to violate the double-blind principle of reviewing, for reproducing experimental results, source codes (with detailed README) will be released if this work is accepted.

---

> > > ### Author Response · Authors · 2022-11-22
> > > **Are raised questions solved?**
> > >
> > > Dear Reviewer vgUK,
> > >
> > > We have provided responses to your follow-up questions, in particular the difference between Herding and our method Moderate-DS. Are there unclear explanations? We could further clarify them.
> > >
> > > Best,
> > > Authors

---

> > > ### Author Response · Authors · 2022-11-26
> > > **Are raised questions solved?**
> > >
> > > Dear Reviewer vgUK,
> > >
> > > Your comments are insightful and constructive. We have tried our best to address all your mentioned concerns. Additionally, as suggested, we have supplemented a series of experiments to strengthen this work.
> > >
> > > After our response, are you still relatively negative about this work? Are there unclear explanations? We can carefully address them.
> > >
> > > Best,
> > > Authors

---

> > > ### Author Response · Authors · 2022-12-01
> > > **Clarifications**
> > >
> > > Dear Reviewer vgUK:
> > >
> > > Your comments are valuable to enhance this work. We have tried our best to address your mentioned concerns. As suggested, a series of experiments are supplemented.
> > >
> > > After our response, do you need any more clarification?
> > >
> > > Best,
> > > Authors

---

> ### Author Response · Authors · 2022-12-07
> **Looking forward to your reply**
>
> Dear Reviewer vgUK:
>
> Thanks for your efforts in reviewing. We are still looking forward to your reply. As the deadline for the discussion is approaching, we tend to ask again whether there are unclear explanations.
>
> We are highly encouraged if your concerns have been addressed. On the contrary, if you need any more clarification, we can provide it as soon as possible, before the discussion deadline.
>
> Best wishes,
> Authors

---

> ### Author Response · Authors · 2022-12-09
> **Still Looking forward to your reply**
>
> Dear Reviewer vgUK:
>
> Thanks for your efforts in reviewing. We have carefully provided detailed responses to your concerns, in particular (1) the differences between our method and baselines, and (2) a series of experiments. As the deadline for the discussion is approaching (**Dec 12**), we tend to ask whether there are unclear explanations.
>
> We are highly encouraged if the concerns have been well addressed. If any more clarification is needed, we can provide it as soon as possible, before the discussion deadline.
>
> Best wishes,
> Authors

---

> ### Author Response · Authors · 2022-12-11
> **Discussion deadline is approaching**
>
> Dear Reviewer vgUK:
>
> Thanks for your efforts in reviewing. We have carefully provided detailed responses to your concerns, in particular (1) the differences between our method and baselines, and (2) a series of experiments as suggested and required.
>
> We are highly encouraged if your concerns have been well addressed. We authors appreciate your reply and thank you again for your efforts in reviewing.
>
> Best wishes,
> Authors

---

### Official Review · Reviewer_GiR4 · 2022-10-24

**Confidence:** 4
**Correctness:** 3
**Technical Novelty And Significance:** 2
**Empirical Novelty And Significance:** 2
**Recommendation:** 6

**Clarity, Quality, Novelty And Reproducibility:**

The paper is clearly written. The motivation for the moderate coreset is reasonable, and the proposed method achieves overall good accuracy. As far as I know, the method to select median distances from the class center is novel. As the method is simple and easy to implement, I expect the results can be reproduced.

**Strength And Weaknesses:**

Strength

- The method performs more robustly compared to other baseline data selection methods under various conditions, including image corruption, label noise, and adversarial attacks.
- The method is simple and easy to implement.
- The paper is clearly written and easy to follow.

Weakness

- Although the overall accuracy is good, the improvement from the baselines looks a bit marginal. Also, the improvement from the baseline seems smaller for higher levels of corruption and label noises.
- I am not sure if the simple average of the features can be a robust representation of the class. If there are outlier samples, then simple averaging will be significantly affected and can be apart from the actual class center of when there are no outliers. Since the proposed moderest coreset is selected based on the distance computed from the centers, which are the simple average of features, it can also be affected.

Other things

- Is the good data selection strategy common among different model architectures? Or is it a property of data itself? What will be the effect of different choices of "well-trained deep model" for representation extraction? How will the median samples and the model performance change?
- Is the good data selection strategy common among different tasks? Or the good coresets differ across tasks? It would be nice if a discussion on the potential extension to general tasks beyond classification could be included.
- How will the method perform for different backbone architectures (e.g. transformer)?
- Another potential advantage of the proposed method could be a better generalization to unseen domains. It would be interesting to see cross-domain evaluation.
- Additional analysis can be done by varying the selection range, e.g. Q1, Q3, in addition to the current Q2.
- In Figure 2, how does the method work for a smaller selection ratio under 60%?
- In Figure 5, how does the method work for more severe corruption (>20%)
- In Table 3, how does the method work for more severe label noise? (>30%)
- I am not sure about the purpose of Figure 4. I guess a comparison with other sampling strategies (Figure 3) with different score computation methods (GraNd-score, EL2N-score) will show the effect of the proposed median-based selection.


**Summary Of The Paper:**

The paper proposes a data selection method for data-efficient deep learning. To make the model robust under various conditions, such as different levels of corruption and label noise, it propose the moderest coreset. Specifically, it first computes the class centers by averaging features of the corresponding samples, and then define the coreset as the samples that have median distances to the center. Experiments show that the method achieves overall good performance for various conditions, including different levels of image corruption and label noise, and adversarial attacks.

**Summary Of The Review:**

In summary, the proposed method is simple but works robustly against various scenarios. However, I think the improvement from the baseline is a bit marginal, and not sure about the significance of the contribution.

---

> ### Author Response · Authors · 2022-11-17
> **Response to Reviewer GiR4 (1/3)**
>
> Thanks for your comments. We address your concerns as follows.
>
> > **Q1**: Although the overall accuracy is good, the improvement from the baselines looks a bit marginal. Also, the improvement from the baseline seems smaller for higher levels of corruption and label noises.
>
> **A1**: Thanks for your comment. The main reason for some marginal improvements from the baselines is that we simply apply the proposed concept of the moderate coreset based on deep representations, considering computational efficiency. The improvement from the baselines is also large sometimes, e.g., in experiments on unseen network structure generalization. Besides, if we apply the moderate-coreset concept to other methods, e.g., GraNd-score and EL2N-score, the improvement is significant (Figure 4). These consistent improvements across different situations support our claims well.
>
> >**Q2**: I am not sure if the simple average of the features can be a robust representation of the class.
>
> **A2**: We understand your concern. The outliers may influence the representation of the class.  However, both the subsequent median operation and moderate-coreset construction can improve the robustness against outliers. Therefore, compared with baselines, our method Moderate-DS can achieve nice performance when there are outliers, e.g., corrupted images and mislabeled data.
>
> There are also some advanced works on robust mean estimation, such as [b1,b2]. However, the complexity of these algorithms is high. We hence simply use the empirical mean for computational efficiency.
>
>
> >**Q3**: Experiments with different model architectures.
>
> **A3**: Thanks for your comment. We employ more network architectures to verify the effectiveness of our method, i.e., VGG-16  (abbreviated as V) [d1] and ShuffleNetv2  (abbreviated as S) [d2], which are popularly used. We conduct experiments on Tiny-ImageNet. As did in the main paper, we consider more challenging cases of small selection ratios (i.e., 20% and 30%) and employ the following settings:
> - Experiments with original datasets
> - Experiments with 20% corrupted images;
> - Experiments with 20% mislabeled data;
> - Experiments with adversarial examples by PGD attacks.
>
> As the reviewer Ergu has the same question, we put the results into the following Tables d-1. d-2, d-3, and d-4 in the response to the reviewer Ergu, which confirm the effectiveness of the proposed method. The experiments are also provided in Appendix C.1 of the revised paper.
>
> >**Q4**: It would be nice if a discussion on the potential extension to general tasks beyond classification could be included.
>
> **A4**: Beyond classification, the concept of the moderate coreset can be applied to object objection and image segmentation. For example, prior work [b3] exploits the uncertainty-adjusted term frequency-inverse document frequency (TF-IDF) to construct the coreset for object objection. The concept of the moderate coreset can be used, where data points whose TF-IDF are close to the median of all TF-IDF are selected. Besides, the work [b4] employs the entropy measurement to select the image regions with higher uncertainty. Similarly, the moderate-coreset concept can be applied to the selection.

---

> > ### Author Response · Authors · 2022-11-17
> > **Response to Reviewer GiR4 (2/3)**
> >
> > >**Q5**: How will the method perform for different backbone architectures (e.g. transformer)?
> >
> > **A5**: Thanks for your advice. We exploit Transformer for experiments with different backbone architectures. The implementation of Transformer is based on the repository (https://github.com/kentaroy47/vision-transformers-cifar10), where ViT small is used. We conduct experiments on simulated CIFAR-100. Following the main paper, we consider: (1) experiments with original datasets (C); (2) experiments with 20% corrupted images (C+C); (3) experiments with 20% mislabeled data (C+M); (4) experiments with adversarial examples (C+A). We consider a more challenging setting, where the selection ratio is 20%.  We provide results in Table b-1, which are also supplemented in Appendix C.2.
> >
> > | Methods             |         C          |        C+C         |        C+M         |        C+A         | Ave. rank |
> > | :------------------ | :----------------: | :----------------: | :----------------: | :----------------: | :-------: |
> > | Random              |   33.80$\pm$0.54   |   30.89$\pm$0.61   |   26.67$\pm$0.54   |   31.01$\pm$0.45   |    3.3    |
> > | Herding             |   32.16$\pm$0.37   |   31.75$\pm$0.22   | **32.27$\pm$0.53** |   31.28$\pm$0.66   |    2.5    |
> > | Forgetting          |   33.52$\pm$0.73   |   24.45$\pm$0.29   |   26.24$\pm$1.07   |   28.26$\pm$1.95   |    5.3    |
> > | GraNd-score         |   22.49$\pm$0.47   |   18.40$\pm$0.11   |   22.13$\pm$0.90   |   19.27$\pm$1.27   |    8.0    |
> > | EL2N-score          |   26.15$\pm$0.21   |   23.27$\pm$0.68   |   24.37$\pm$0.82   |   22.06$\pm$1.68   |    7.0    |
> > | Optimization-based  |   31.84$\pm$0.63   |   30.12$\pm$0.73   |   30.11$\pm$0.70   |   29.36$\pm$0.75   |    4.5    |
> > | Self-sup.-selection |   33.35$\pm$0.31   |   30.72$\pm$0.90   |   29.16$\pm$0.27   |   28.49$\pm$0.56   |    4.3    |
> > | Moderate-DS         | **34.43$\pm$0.32** | **32.73$\pm$0.35** |   31.86$\pm$0.49   | **32.61$\pm$0.40** |  **1.3**  |
> >
> > Table b-1: Experimental results on simulated CIFAR-100 with Transformer. The selection ratio is set to 20%.
> >
> > >**Q6**: It would be interesting to see cross-domain evaluation.
> >
> > **A6**: As suggested, we employ a cross-domain evaluation to verify the effectiveness of our method. Specifically, we employ the dataset Office-Home [b5], which is popularly used in domain adaption. We train the deep model and perform data selection on the "Product" domain, and test the model on the "Clipart" domain. The selection ratios are set to 20%, 30%, and 40% respectively. We provide experimental results in Table b-2, which demonstrate the superiority of the proposed method under cross-domain settings.
> >
> > | Methods             |        20%         |        30%         |        40%         |
> > | :------------------ | :----------------: | :----------------: | :----------------: |
> > | Random              | **37.60$\pm$1.49** |   39.06$\pm$0.92   |   40.29$\pm$1.45   |
> > | Herding             |   34.92$\pm$0.66   |   37.25$\pm$0.88   |   39.89$\pm$0.71   |
> > | Forgetting          |   30.95$\pm$0.91   |   33.94$\pm$1.06   |   37.62$\pm$0.84   |
> > | GraNd-score         |   33.65$\pm$0.47   |   38.15$\pm$0.85   |   39.62$\pm$0.70   |
> > | EL2N-score          |   29.92$\pm$1.26   |   34.18$\pm$0.90   |   37.77$\pm$0.59   |
> > | Optimization-based  |   32.68$\pm$0.47   |   34.17$\pm$0.93   |   38.65$\pm$1.32   |
> > | Self-sup.-selection |   35.84$\pm$0.89   |   38.06$\pm$0.24   |   40.46$\pm$0.93   |
> > | Moderate-DS         |   36.24$\pm$0.25   | **39.84$\pm$0.37** | **41.32$\pm$0.68** |
> >
> > Table b-2: Experimental results about cross-domain evaluations.
> >
> > >**Q7**: In Figure 2, how does the method work for a smaller selection ratio under 60%?
> >
> > **A7**: As the time costs of experiments on ImageNet-1k are huge, we here provide the results when the selection ratio is 50%. The results are shown in Table b-3, where our Moderate-DS achieves the best performance.
> >
> > | Random | Herding | Forgetting | GraNd-score | EL2N-score | Self-sup.-selection | Moderate-DS |
> > | :----: | :-----: | :--------: | :---------: | :--------: | :-----------------: | :---------: |
> > | 84.81  |  85.36  |   85.60    |    85.26    |   85.13    |        85.06        |  **85.78**  |
> >
> > Table b-3: Experiments on ImageNet-1k with the 50% selection ratio.

---

> > > ### Author Response · Authors · 2022-11-17
> > > **Response to Reviewer GiR4 (3/3)**
> > >
> > > >**Q8**: In Figure 5, how does the method work for more severe corruption (>20%)?
> > >
> > > **A8**: To address your concern and make experiments more convincing, the rate of corruption is improved to 30%. We conduct experiments on both simulated CIFAR-100 and Tiny-ImageNet. Experimental results are provided in Tables b-4 to verify the effectiveness of our method.
> > >
> > > | Methods             | 20% (C) | 30% (C) | 20% (T) | 30% (T) | Ave. rank |
> > > | :------------------ | :--: | :--: | :--: | :--: | :--: |
> > > | Random              | 39.67$\pm$1.72 | 48.89$\pm$1.68 | 17.34$\pm$1.28 | 23.14$\pm$0.96 | 4.3 |
> > > | Herding             | **43.25$\pm$1.11** | **53.24$\pm$1.06** | 12.04$\pm$0.66 | 18.03$\pm$0.58 | 4.5 |
> > > | Forgetting          | 26.59$\pm$1.29 | 39.67$\pm$1.95 | 13.68$\pm$0.45 | 21.10$\pm$0.32 | 6.8 |
> > > | GraNd-score         | 31.96$\pm$1.58 |   45.19$\pm$1.61   | 14.85$\pm$0.39 | 21.06$\pm$0.35 | 6.0 |
> > > | EL2N-score          | 20.53$\pm$0.89 | 26.62$\pm$1.13 | 16.60$\pm$0.48 | 23.90$\pm$0.15 | 6.3 |
> > > | Optimization-based  | 33.14$\pm$0.62 | 40.86$\pm$2.30 | 18.55$\pm$0.16 | 25.73$\pm$0.82 | 4.3 |
> > > | Self-sup.-selection | 41.53$\pm$0.83 | 52.52$\pm$1.38 | 20.09$\pm$0.40 | 26.07$\pm$0.26 | 2.5 |
> > > | Moderate-DS         | 42.58$\pm$0.64 | 53.14$\pm$0.89 | **20.65$\pm$0.12** | **27.32$\pm$0.36** | **1.5** |
> > >
> > > Table b-4: Experimental results on simulated CIFAR-100 (C) and Tiny-ImageNet (T) with 30% corrupted images.
> > >
> > >
> > > >**Q9**: In Table 3, how does the method work for more severe label noise? (>30%)?
> > >
> > > **A9**: Thanks for your comment. We set the label noise rate to 35% to verify the effectiveness of the proposed method. As can be seen, in this case, our method still works well, especially for the experiments on Tiny-ImageNet.
> > >
> > >
> > >
> > > | Methods             |      20% (C)       |      30% (C)       |      20% (T)       |      30% (T)       | Ave. rank |
> > > | :------------------ | :----------------: | :----------------: | :----------------: | :----------------: | :-------: |
> > > | Random              |   24.51$\pm$1.34   |   32.26$\pm$0.81   |   14.64$\pm$0.29   |   19.41$\pm$0.45   |    4.8    |
> > > | Herding             |   29.42$\pm$1.54   |   37.50$\pm$2.12   |   15.14$\pm$0.45   |   20.19$\pm$0.45   |  **2.0**  |
> > > | Forgetting          | **29.48$\pm$1.98** | **38.01$\pm$2.21** |   11.25$\pm$0.90   |   17.07$\pm$0.66   |    3.8    |
> > > | GraNd-score         |   23.03$\pm$1.05   |   34.83$\pm$2.01   |   13.68$\pm$0.46   |   19.51$\pm$0.45   |    5.0    |
> > > | EL2N-score          |   21.95$\pm$1.08   |   31.63$\pm$2.84   |   10.11$\pm$0.25   |   13.69$\pm$0.32   |    8.0    |
> > > | Optimization-based  |   26.77$\pm$0.15   |   35.63$\pm$0.92   |   12.37$\pm$0.68   |   18.52$\pm$0.90   |    4.5    |
> > > | Self-sup.-selection |   23.12$\pm$1.47   |   34.85$\pm$0.68   |   11.23$\pm$0.32   |   17.76$\pm$0.69   |    6.0    |
> > > | Moderate-DS         |   28.45$\pm$0.53   |   36.55$\pm$1.26   | **15.27$\pm$0.38** | **20.33$\pm$0.28** |  **2.0**  |
> > >
> > > Table b-5: Experimental results on simulated CIFAR-100 (C) and Tiny-ImageNet (T) with label noise, where the noise rate is 35%.
> > >
> > > > **Q10**: On the purpose of Figure 4.
> > >
> > > **A10**: Yes, your understanding is right. The comparison in Figure 4 shows that the concept of the moderate coreset can be applied to other methods (i.e., GraNd-score and EL2N-score) for performance improvement.
> > >
> > > ----
> > > [b1] Kevin A. Lai et al. Agnostic Estimation of Mean and Covariance. FOCS 2016
> > > [b2] Ilias Diakonikolas et al. Robust Sparse Mean Estimation via Sum of Squares. COLT 2022
> > > [b3] Jiaqi Fan et al. Data Subset Selection for Object Detection. Pre-registration workshop NeurIPS 2020
> > > [b4] Deepthi Sreenivasaiah et al. MEAL: Manifold Embedding-based Active Learning. arXiv preprint: 2106.11858
> > > [b5] Hemanth Venkateswara et al. Deep Hashing Network for Unsupervised Domain Adaptation. CVPR 2017

---

> ### Author Response · Authors · 2022-11-22
> **Further Discussion**
>
> Dear Reviewer GiR4,
>
> Many thanks for your valuable comments. They will enhance this paper without any doubt. We are open to further discussions on the work.
>
> Best,
> Authors

---

> ### Author Response · Authors · 2022-11-26
> **Thanks for your comments**
>
> Dear Reviewer GiR4,
>
> Thanks for your insightful and valuable comments. We have provided responses to address the concerns and supplemented experiments carefully. We are looking forward to hearing your further opinion on our vision paper. Thank you very much!
>
>
> Best,
> Authors

---

> > ### Comment · Reviewer_GiR4 · 2022-11-28
> > **Thank you for the answers**
> >
> > I thank the authors for the detailed answers. They have addressed most of my concerns.

---

### Official Review · Reviewer_vuQC · 2022-10-25

**Confidence:** 5
**Correctness:** 4
**Technical Novelty And Significance:** 2
**Empirical Novelty And Significance:** 3
**Recommendation:** 8

**Clarity, Quality, Novelty And Reproducibility:**

Clarity. There are some unclear points in this paper, which need to be addressed carefully.

(1) The proposed moderate sets exploit the data points with scores close to the score median. It may be more common that we regard the mean/location as a proxy of a distribution. Why did this paper exploit the median? There is no enough discussion on this problem.

(2) The baseline “Self-sup.-selection” has a similar claim that we should select easy examples for small datasets and hard examples for large datasets, which could be mentioned in the discussions in Section 2.2.

(3) The mutual information estimator is critical for the justification. One suggestion is to provide more technical details about it.

(4) More baselines in data selection, e.g., [1-3], can be added to make the results more convincing.

[1] Cody Coleman et al. Selection via Proxy: Efficient Data Selection for Deep Learning. ICLR 2020.
[2] Kristof Meding et al. Trivial or Impossible—Dichotomous Data Difficulty Masks Model Differences (on ImageNet and Beyond). ICLR 2022.
[3] Vitaly Feldman and Chiyuan Zhang. What Neural Networks Memorize and Why: Discovering the Long Tail via Influence Estimation. NeurIPS 2020.

(5) In practice, a dataset always is corrupted by various sources at the same time. Although the paper has presented the proposed method works well in the cases of clean data, corrupted images, and adversarial examples respectively. It will be interesting to see how different methods work when all these factors simultaneously exist. The setting may be more realistic.

(6) Apart from the baseline Herding, the other baseline methods mainly stress that those “difficult” examples are more helpful. For example, Forgetting selects the examples that are difficult to be memorized. In fact, these methods can also be revised to select “easy” examples. It is interesting to see how these methods work after such revisions.

(7) Figure 4 shows that the baselines EL2N and Grand can be improved with the concept of moderate sets. Could the paper provide more such evidence to strengthen the contributions of this paper?

Quality. The quality of this paper is great. Overall, the writing is good. The descriptions of the motivation, research problem, and solutions are clear, following convincing experimental results.

Novelty. The idea is novel to me and is much potential for future research.

Reproducibility. The descriptions of experimental settings are detailed. The reproducibility is satisfactory.


**Strength And Weaknesses:**

Pros:
- The motivation of this paper is clear and strong. Although data selection methods have been widely studied in different research topics, there may be no method that is designed to cope with a variety of scenarios at the same time. The paper also gives careful analyses of the vulnerability of existing methods to scenario changes.
- The proposed method is simple but effective. The method Moderate-DS works with extracted deep representations, which avoids model retraining and access to network structures. The advantages may make it easier to apply in practice.
- The experimental results are promising. Although the proposed method does not perform the best in all cases, the results show that it can achieve the best performance in most cases and is competitive in others. Moreover, the proposed concept can be applied to other methods to bring practical improvements.
- The writing of this paper is overall great.

Cons:
- Technical contributions of this paper seems trivial. Although this paper provides insights into the research community, the method implementation is not very technical.
- More comparison methods can be added to enhance this paper.


**Summary Of The Paper:**

This paper studies data selection to construct a subset of full data, which is an important research topic in data-efficient deep learning. The authors piont out that prior works on data selection are always specially designed for certain cases, which makes it hard to apply them in practice, since realistic scenes are ever-changing and mismatch pre-defined ones. To address the issue, the concept of moderate sets and a new algorithm are proposed, where the data located in the middle of the data distribution is selected. Extensive experiments on multiple tasks demonstrate the effectiveness of the proposed concept and method. A strong baseline is created for future research.

**Summary Of The Review:**

This paper focuses on data selection to boost data-efficient learning. The motivation of this paper is strong. To address the issues of prior work, the concept of moderate, and a simple and effective method are proposed. Although the paper does not provide theoretical proofs for the method, both justifications from representation learning/information bottleneck and empirical evidence are provided. Besides, the proposed method is simple but with impressive results. The reviewer appreciates the inspiration this paper provides to the research community and its value in practical applications. Therefore, it is recommended for acceptance.

---

> ### Author Response · Authors · 2022-11-17
> **Response to Reviewer vuQC (1/2)**
>
> Thanks for your comments. We address your concerns as follows.
>
> > **Q1**:  Why did this paper exploit the median but not the mean?
>
> **A1**: Although both the median and mean can be regarded as a proxy of a distribution, the median is more robust to diverse scenes. For example, the tail of a distribution may shift the mean heavily towards the tail. The calculation of the mean is much easier to be influenced by outliers. Similar views are also claimed in robust statistics such as [a1,a2].
>
> > **Q2**: The baseline “Self-sup.-selection” has a similar claim, which could be mentioned in the discussions in Section 2.2.
>
> **A2**: Thanks for pointing out this. We have revised the discussions in Section 2.2 as suggested.
>
> > **Q3**: More technical details about the mutual information estimator.
>
> **A3**: We supplement the technical details of the used mutual information estimator in Appendix D of the revised paper.
>
> > **Q4**: More baselines in data selection can be added to make the results more convincing.
>
> **A4**: As suggested, we exploit the mentioned baselines for comparison. As [a5] does not provide public source codes, we involve the other two baselines. Specifically, we conduct experiments on Tiny-ImageNet. We name the comparison methods as SVP [a4] and Memorization [a6]. As did in the main paper, we consider various realistic situations:
> - Experiments with original datasets (see Table a-1);
> - Experiments with 20% corrupted images (see Table a-2);
> - Experiments with 20% mislabeled data (see Table a-3);
> - Experiments with adversarial examples by PGD attacks (see Table a-4).
>
> | Methods            |        20%         |        30%         |        40%         |
> | :----------------- | :----------------: | :----------------: | :----------------: |
> | SVP                |   24.05$\pm$1.02   |   28.29$\pm$0.73   |   31.77$\pm$0.53   |
> | Memorization       |   23.17$\pm$0.19   |   28.60$\pm$0.15   |   32.37$\pm$0.25   |
> | Moderate-DS (ours) | **25.29$\pm$0.38** | **30.57$\pm$0.20** | **34.81$\pm$0.51** |
>
> Table a-1: Experiments with original datasets.
>
> | Methods            |        20%         |        30%         |        40%         |
> | :----------------- | :----------------: | :----------------: | :----------------: |
> | SVP                |   21.33$\pm$0.09   |   26.27$\pm$1.83   |   31.39$\pm$0.50   |
> | Memorization       |   20.90$\pm$0.64   |   26.24$\pm$0.64   |   31.95$\pm$0.67   |
> | Moderate-DS (ours) | **23.27$\pm$0.33** | **29.06$\pm$0.36** | **33.48$\pm$0.11** |
>
> Table a-2: Experiments with 20% corrupted images.
>
> | Methods            |        20%         |        30%         |        40%         |
> | :----------------- | :----------------: | :----------------: | :----------------: |
> | SVP                |   15.39$\pm$0.98   |   22.83$\pm$0.72   |   27.94$\pm$0.90   |
> | Memorization       |   18.06$\pm$0.82   |   22.93$\pm$1.60   |   28.17$\pm$1.51   |
> | Moderate-DS (ours) | **19.64$\pm$0.40** | **24.96$\pm$0.30** | **29.56$\pm$0.21** |
>
> Table a-3: Experiments with 20% mislabeled data.
>
> | Methods            |        20%         |        30%         |        40%         |
> | :----------------- | :----------------: | :----------------: | :----------------: |
> | SVP                |   20.39$\pm$0.52   |   26.38$\pm$0.74   |   31.30$\pm$0.67   |
> | Memorization       |   20.82$\pm$0.90   |   26.75$\pm$0.84   |   31.45$\pm$0.25   |
> | Moderate-DS (ours) | **21.81$\pm$0.37** | **27.11$\pm$0.20** | **32.03$\pm$0.17** |
>
> Table a-4: Experiments with adversarial examples (PGD attacks).
>
> > **Q5**: How do different methods work when all these factors simultaneously exist?
>
> **A5**: Thanks for pointing out the realistic case. We take your advice and supplement empirical evaluations. Specifically, we simulate the realistic case by injecting 10% corrupted images, 10% mislabeled data, and 10% adversarial examples (PGD attacks) into original datasets. We conduct experiments with simulated Tiny-ImageNet. Empirical results are provided in Table a-5, which are also supplemented in Appendix C.5 of the revised paper.
>
> | Methods     | 20% | 30% | 40% |
> | :---------- | :-----: | :-----: | :-----: |
> | Random      | 19.08$\pm$0.56 | 24.69$\pm$0.59 | 28.72$\pm$0.89 |
> | Herding     | 19.08$\pm$0.23 | 24.63$\pm$0.19 | 28.65$\pm$0.16 |
> | Forgetting  | 15.96$\pm$0.21 | 23.79$\pm$0.17 | 29.23$\pm$0.19 |
> | GraNd-score | 18.91$\pm$0.34 | 25.22$\pm$0.15 | 28.57$\pm$0.45 |
> | EL2N-score |15.36$\pm$0.25|21.18$\pm$0.21|25.83$\pm$0.52|
> | Optimization-based |19.05$\pm$1.00|24.93$\pm$0.46|29.09$\pm$1.02|
> | Self-sup.-selection |17.19$\pm$0.82|22.55$\pm$0.29|28.45$\pm$0.28|
> |Moderate-DS|**20.91$\pm$0.36**|**26.24$\pm$0.19**|**30.31$\pm$0.10**|
>
> Table a-5: Experiments on simulated Tiny-ImageNet, where corrupted images, mislabeled data, and adversarial examples are injected into the original dataset.

---

> > ### Author Response · Authors · 2022-11-17
> > **Response to Reviewer vuQC (2/2)**
> >
> > > **Q6**: It is interesting to see how these methods work after the mentioned revisions.
> >
> > **A6**: We revise them as suggested. Here, the method "A" after revised is named "A (R)". We conduct experiments on original Tiny-ImageNet (abbreviated as T) and Tiny-ImageNet with 20% corrupted images (abbreviated as T+C). Empirical results are provided in Table a-6. The effectiveness of our method can be verified.
> >
> > | Methods                 |      20% (T)       |     20% (T+C)      |
> > | :---------------------- | :----------------: | :----------------: |
> > | Forgetting (R)          |   23.03$\pm$0.23   |   19.95$\pm$0.18   |
> > | GraNd-score (R)         |   23.33$\pm$0.34   |   22.36$\pm$0.39   |
> > | EL2N-score (R)          | **25.79$\pm$0.21** |   22.99$\pm$0.19   |
> > | Optimization-based (R)  |   24.03$\pm$0.13   |   22.16$\pm$0.93   |
> > | Self-sup.-selection (R) |   23.70$\pm$0.23   |   15.98$\pm$0.43   |
> > | Moderate-DS             |   25.29$\pm$0.38   | **23.27$\pm$0.33** |
> >
> > Table a-6: Experiments on Tiny-ImageNet and corrupted Tiny-ImageNet. The baselines are revised to address those "easy" examples.
> >
> > >**Q7**: Please provide more evidence to strengthen the contributions of this paper.
> >
> > **A7**: We apply the concept of the proposed moderate coreset to the method Forgetting. Specifically, we conduct experiments on CIFAR-100. The boosted method with the moderate coreset is called Forgetting+. We provide results in Table a-7 to justify our claim.
> >
> > | Methods     |        20%         |        30%         |
> > | :---------- | :----------------: | :----------------: |
> > | Forgetting  |   35.57$\pm$1.40   |   49.83$\pm$0.91   |
> > | Forgetting+ | **41.24$\pm$0.96** | **55.73$\pm$0.73** |
> >
> > Table a-7: Experiments for improving Forgetting with the concept of the moderate coreset.
> >
> > ----
> > [a1] Peter J Rousseeuw. Tutorial to robust statistics. Journal of chemometrics, 1991.
> > [a2] Daniel  Gervini. Robust functional estimation using the median and spherical principal components. Biometrika, 2008.
> > [a3] Mohamed Ishmael Belghazi et al. Mine: mutual information neural estimation. ICML 2018.
> > [a4] Cody Coleman et al. Selection via proxy: efficient data selection for deep learning. ICLR 2020.
> > [a5]  Kristof Meding et al. Trivial or impossible—dichotomous data difficulty masks model differences (on ImageNet and beyond). ICLR 2022.
> > [a6] Vitaly Feldman and Chiyuan Zhang. What neural networks memorize and why: discovering the long tail via influence estimation. NeurIPS 2020.

---

> > > ### Comment · Reviewer_vuQC · 2022-11-22
> > > **All my concerns are well addressed**
> > >
> > > Thank you for the detailed response, and it has well addressed all my concerns. Therefore, I will keep my score and recommend this paper for publication.

---

> > > > ### Author Response · Authors · 2022-11-22
> > > > **Thanks for your response**
> > > >
> > > > Dear Reviewer vuQC,
> > > >
> > > > Thanks a lot for your response. Your comments are constructive to improve this work.
> > > >
> > > > Best,
> > > > Authors

---

> ### Author Response · Authors · 2022-11-22
> **Further Discussion**
>
> Dear Reviewer vuQC,
>
> Thanks a lot for your valuable comments. We have provided responses to address your concerns. Are there unclear explanations here? We could further clarify them.
>
> Best,
> Authors

---

### Author Response · Authors · 2022-11-17
**General Response**

We thank the reviewers for their insightful and constructive reviews of our manuscript. Besides, we are highly encouraged to hear that the reviewers found that the core idea is interesting and novel (Reviewers vuQC, GiR4, and Ergu), the proposed method is easy to implement (Reviewers vuQC, GiR4, and Ergu), experiments are overall convincing (Reviewers vuQC and Ergu), and writing is great (Reviewers vuQC, GiR4, vgUK, and Ergu). Based on the reviews, we provide a general response here to the points raised by multiple reviewers, and individual responses below to address each reviewer’s concerns. Importantly,

(1) Regarding questions about experiments, we have addressed the concern as follows:

- For Reviewer vuQC, we supplement: (a) the experiments with mentioned baselines; (b) experiments with simultaneous corrupted/mislabeled/adversarial examples; (c) experiments when baselines work after revisions.
- For Reviewer GiR4, we add: (a) experiments with different model architectures; (b) experiments with more corrupted images; (c) experiments with more mislabeled data; (d) experiments in the cross-domain setting.
- For Reviewer vgUK, we add: (a) quantitative time comparison; (b) experiments about more selection ratios when mislabeled data occur.
- For Reviewer Ergu, we supplement: (a) experiments with model architectures; (b) how the proposed concept and baselines benefit each other.

(2) Regarding questions about the idea and technical details, we have addressed the concern as follows:

- For Reviewer vuQC, we explain the use of the median and add more descriptions to enhance this paper.
- For Reviewer GiR4, we add explanations on performance improvement and calculations of class centers.
- For Reviewer vgUK, we address the concerns about the baseline Herding and stress the difference between Herding and our method from both philosophical and technical aspects.



We have revised our draft according to your valuable comments. Major revisions are highlighted in blue. We sincerely thank all the reviewers. Please feel free to let us know if further details/explanations would be helpful.

Best,
Authors

---

### Decision · Program_Chairs · 2023-01-20

**Decision:**

Accept: poster

**Justification For Why Not Higher Score:**

The paper shows an interesting and simple model, easy to implement and a strong baseline however a more sophisticated contributions should be necessary for a spotlight.

**Justification For Why Not Lower Score:**

The paper shows an interesting paper that can not be rejected.

**Metareview: Summary, Strengths And Weaknesses:**

# Summary
This paper focuses on data selection to construct a subset of full data, which is an important research topic in data-efficient deep learning. To address the issue of designing algorithms for certain cases, the concept of moderate sets and a new algorithm are proposed.  Specifically, it first computes the class centers by averaging features of the corresponding samples, and then define the coreset as the samples that have median distances to the center. Extensive experiments on multiple tasks demonstrate the effectiveness of the proposed concept and method.
# Strengths:
- the core idea is interesting
- the proposed method is easy to implement
- experiments are overall convincing (Reviewers vuQC and Ergu)
- writing is great
- a strong baseline is created for future research.
# Weaknesses:
- Technical contributions of this paper seems trivial. Although this paper provides insights into the research community, the method implementation is not very technical.
- Only tested in one neural network architecture


**Note From Pc:**

if the above contains the word "oral" or "spotlight" please see: "oral" presentation means -> notable-top-5% and "spotlight" means -> notable-top-25%. As stated in our emails, we are disassociating presentation type from AC recommendations